# Graphlets as Building Blocks for Structural Vocabulary in Graph Foundation Models

## Abstract

Foundation models excel at language, where sentences become tokens, and vision, where images become pixels, because both reduce to discrete symbols on a shared, fixed grid. Knowledge Graphs share the discreteness, but not the geometry. Their entities and relations are discrete symbols, yet their arrangement is relational and lacks a common, fixed grid. Knowledge Graphs (KGs) share the discreteness, but not the geometry. They form irregular, non-Euclidean topologies whose local neighborhoods differ from graph to graph. Therefore, Knowledge Graph Foundation Models (KGFMs) rely on identifying structural invariances to produce transferable representations. Without a universal token set, KGFMs are limited in their ability to transfer representations across unseen KGs. We close this gap by treating graphlets, small connected graphs, as structural tokens that recur in heterogeneous KGs. In this paper, We introduce a model-agnostic framework based on a vocabulary of graphlets that mines a KG between relations via pattern matching. In particular, we considered closed and open 2- and 3-path, and star graphlets, to obtain robust invariances. The framework is evaluated on 51 KGs from a wide range of domains, for zero-shot inductive and transductive link prediction. Experiments show that adding simple graphlets to the vocabulary yields models that outperform prior KGFMs. Our code is available at: https://anonymous.4open.science/r/ultra-augmentations/

## 1 Introduction

Recently, Large language Models (LLMs), have garnered significant attention for their remarkable natural language understanding capabilities (Bommasani, 2021; Zhao et al.; Wei et al., 2022). These models are pretrained on massive corpora of diverse text data (Chang et al., 2024), allowing them to learn not only the syntax and grammar of language but also the semantics and contextual usage of words and phrases. Despite differences in architecture (e.g., transformer-based, decoder-only, encoder-decoder), all LLMs operate on tokens; basic units of text that may be whole words or subwords. During training, the models construct a universal vocabulary of tokens. This token-based processing enables LLMs to generalize effectively to new words by breaking them down into familiar token components. In turn, complete sentences can be reconstructed from these tokens. As a result, LLMs achieve strong generalization across languages (Lin et al., 2021), domains (Pan et al., 2024), and (Brown et al., 2020; Wang et al., 2022). The (unstructured) textual data can be saved as tripled-based data, known as Knowledge Graphs (Hogan et al., 2021; Noy et al., 2019; Ehrlinger & Wöß, 2016). Knowledge Graph embeddings (KGEs) are a class of representation learning models specialized for Knowledge Graphs (KGs). They learn entity and relation embeddings based on their labeled identities and the structure of the triples they participate in (Wang et al., 2017). While effective at modeling relational patterns, they are limited in their ability to generalize to unseen entities or relations.

In contrast to LLMs, KGEs do not capture any natural understanding of the labels themselves. As a result, adapting KGEs to new entities or relation types typically requires retraining from scratch on augmented data (Hamaguchi et al., 2017; Teru et al., 2020). To overcome this limitation, recent approaches explore structure-driven generalization in (Liu et al., 2021). One idea is to treat structural patterns in a graph analogously to how LLMs treat tokens in text data. These patterns capture local and global structural invariants independent of specific entity labels or relation types.

**Motivating Example.** Let us consider the three illustrative KGs: Family, Corporate and Scholarly, shown in Figure 1. Despite the fact that the type of relations and entity labels are not the same, their underlying structural topology is the same. This allows for a mapping between their rela-

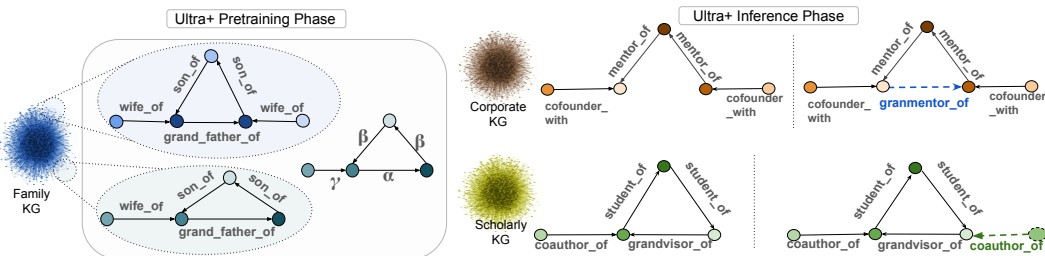

Figure 1: The KGFM model $\text{Ultra}^+$, pretrained on a large collection of KGs, including the Family KG, recognizes the Corporate, and Academic KGs as instances of the same graphlet patterns.

tions (*grand_father_of* ↔ *grandvisor*, *son_of* ↔ *mentor_of*, *wife_of* ↔ *cofounder_with*) enabling structure-level transfer. This core insight forms the basis of KGFMs. Rather than embedding labels, KGFMS (Galkin et al., Huang et al.) learn to reason over vocabulary of relations forming relational subgraphs. Group of ordered relations that co-occur within a specific type of subgraph is referred to as an occurrence of a graphlet. These graphlets form the core of the structural vocabulary used to construct a relation graph, where relations become nodes, and the graphlets define typed edges between them. As shown in Figure 1, a KGFM would detect and save the pattern formed by the cycle $(\gamma, \alpha, \beta, \beta)$ as part of its vocabulary. This learned pattern can then be used to infer missing facts such as the triple *mentor_of*(A. Einstein, ?). However, existing KGFMs have limitations. First, they fail to distinguish between closed and open paths, treating all subgraphs of similar size as equally informative. Second, a single occurrence of a graphlet is often enough to connect involved relations in the relation graph, which may lead to decreased robustness.

**Our Contributions.** The key contributions of our model $\text{Ultra}^+$ are as follows

(i) **Query-based relation graph extraction**: We propose a flexible SPARQL-based extraction method that efficiently identifies informative structures without relying on sparse matrix multiplication (SPMM);

(ii) **Closed and open relations**: We incorporate both closed and open graphlets to capture a wider range of relational patterns;

(iii) **Binary relation**: We represent graphlets as positional binary relations than traditional n-ary relations; and

(iv) **Model-agnostic design**: Ultra+ is modular and can be integrated into any KGFM that uses relation graphs or structural vocabulary, making it adaptable across KGFM architectures.

$\text{Ultra}^+$ substantially strengthens how KGFMs capture and represent complex structural patterns. Its main objective is not to propose a new model architecture, but to boost performance and create more robust KGFMs by enriching their structural vocabulary.

## 2 RELATED WORK

A KGE model learns the vector representations of relations ($\mathbf{r}$) and entities ($\mathbf{E}$) of KGs via parametrized relation-specific transformation functions, $\Phi_{\mathbf{r}} : \mathbf{h} \mapsto \Phi_{\mathbf{r}}(\mathbf{h})$, over the entity embedding space. KGEs can be categorized based on their underlying principles such as geometric transformations, tensor decomposition, deep neural architectures, or foundational graph approaches.

**Geometric and Tensor Decomposition Models.** These models can be grouped into three main types. The first are translational-based models such as $\text{TransE}$ (Bordes et al., 2013) , $\text{TransH}$ (Wang et al., 2014), and $\text{TransR}$ (Lin et al., 2015), which embed entities and relations in real vector space and use identity mappings ($\Phi_{\mathbf{r}} = \mathbf{r}$). $\text{TransH}$ and $\text{TransR}$ add extra relation embeddings, to improve modeling capacity. While effective for link prediction, these models struggle with relational patterns like closed paths. To address these limitations, rotation-based models such as $\text{RotatE}$ (Sun et al., 2019) were introduced. $\text{RotatE}$ represents relations as rotations: $\Phi_{\mathbf{r}}(\mathbf{h}) = e^{i\theta_r}\mathbf{h}$ with $\theta_r \in (-\pi, \pi]^d$.

This shift from real to complex algebra enables better modeling of relational patterns. However, these models are not generalizable to new entities and relations.

**Deep Neural Network Models.** These models leverage deep learning to extract graph representations. A primary category includes Graph Neural Networks (GNNs), especially graph convolutional models that iteratively aggregate information from neighboring nodes. A pioneer work is Relational GNC (Schlichtkrull et al., 2018), an encoder-decoder model. The encoder of R-GCN learned latent embedding of entities, which are passed to the decoder based on DistMult, a tensor decomposition model. However, R-GCNs does not learn relation embeddings. To address this limitation, TransGCN, RotatEGCN (Cai et al., 2019), and ComplexGCN (Zeb et al., 2022) integrate GCNs with geometric KGE models like TransE, RotatE, and ComplEx to jointly learn entity and relation embeddings and capture richer structural semantics. End-to-end trainable GNNs are restricted to a single KG downstream task and cannot generalize to new KGs without retraining.

**Knowledge Graph Foundation Models.** KGFMs overcome these limitations by enabling pretrained GNNs or LLMs to inductively generalize to new KGs in zero or few-shot paradigms (Wang et al., 2025; Liu et al., 2023). ULTRA (Galkin et al., 2023), a KGFM for KG reasoning, constructs a relation graph whose nodes are the relations from the original KG, and edges represent the connections between relations in paths of length two. Leveraging on the invariance of relational structure across datasets, a labeling trick, and conditional representations on both relations and entities, Ultra enhances the generalization of KG reasoning. ULTRAQuery (Galkin et al., 2024) exploits the ability of Ultra in KG reasoning to find potential missing links, and uses non-parametric fuzzy logic operators to answer complex questions. AnyGraph (Xia & Huang, 2024) overcomes the limit of Ultra, by generalizing to in- and cross- domain link prediction, and node and graph classification tasks. The key factor behind the success of existing KGFMs lies in the construction of suitable graph vocabularies (Mao et al., 2024), i.e. basic transferable units that underlie graphs. While models such as Mole-BERT relies on context-aware atom vocabulary (Xia et al., 2023) for molecule graph classification, Ultra and Motif (Huang et al., 2025) rely on paths of length two and motifs to define graph vocabulary, respectively. However, these approaches overlook closed paths, which are prevalent and essential.

In this paper, we extend the Ultra framework by introducing a novel graphlet-based vocabulary for KG reasoning. Unlike Ultra, our framework explicitly encodes cyclic structures, this allows us to capture richer structural patterns beyond simple paths of length two. Moreover, in contrast to Motif, our vocabulary supports higher-order interaction patterns via binary relations within a standard relation graph, rather than relying on n-ary relations in a relation hypergraph. This ensures our relation graph remains a simple KG, preserving compatibility with most established KG processing techniques.

## 3 Preliminaries

### 3.1 Inductive Knowledge Graph Embeddings

We consider a *Knowledge Graph* as a multi-relational directed graph $K = (\mathcal{E}^K, \mathcal{R}^K, \mathcal{T}_+^K)$ where $\mathcal{E}^K, \mathcal{R}^K$, and $\mathcal{T}_+^K$ are the set of nodes (entities), edge labels (relations), and ordered pairs (edges or triples) formed as *relation(head entity, tail entity)* respectively. We refer to the head and tail entity as $h$ and $t$ or $e$ in general, and to the relation as $r$ or $q$. $\mathcal{T}_+^K$ is a subset of $\mathcal{T}^K$ which contains all plausible triples; and $\mathcal{T}_-^K = \mathcal{T}^K \setminus \mathcal{T}_+^K$ is the set of corrupted triples. Since all entities in $\mathcal{E}^K$ are used in constructing $\mathcal{T}_+^K$, corrupted triples (used as negative samples) result from replacing the head or the tail entity of true triples. The set $\mathcal{N}(r,t) = \{h | r(h,t) \in \mathcal{T}^+\}$ is called the neighborhood of $t$.

A *relation graph* is constructed from the KG by examining how relations within a target KG appear collectively in subgraphs. The labels on its edges and nodes originate from subgraph configurations and relations within the target KG. The relations in a relation graph can also be referred to as *meta-relations* (see Section 4.1 for more details).

KGE models pretrained on $K$ are evaluated on a test KG $K_{test} = (\mathcal{E}_{test}^K, \mathcal{R}_{test}^K, \mathcal{T}_{test+}^K)$ to predict missing links. Entities in $\mathcal{E}_{test}^K \setminus \mathcal{E}^K$ and relations in $\mathcal{R}_{test}^K \setminus \mathcal{R}^K$ are called unseen entities and relations, respectively. A KGE model is a *transductive* model if $\mathcal{E}_{test}^K \subseteq \mathcal{E}^K$ and $\mathcal{R}_{test}^K \subseteq \mathcal{R}^K$, an *inductive* model otherwise. *Zero-Shot Link Prediction* (ZSLP) involves evaluating models pretrained on $K$, directly on $K_{test}$. Inductive ZSLP can be categorized into three main tasks: *Relation learning* involving predicting unseen relations (Ind.(r)), *entity learning* focusing on predicting facts involving

unseen entities (Ind.(e)), and *graph transfer* requiring generalization to both unseen entities and relations (Ind.(e,r)).

## 3.2 Knowledge Graph Homomorphism

A key feature of our framework is its ability to detect occurrences of specific subgraph patterns within a KG by leveraging graph homomorphisms for structural matching.

A *KG homomorphism* is a structure preserving mapping between two KGs. It consists of entity and relation mappings to relate entities and relations from one KG to the other. Thus, $\phi : K \to K'$ is a KG homomorphism if there exists two mappings $\eta : \mathcal{E}^K \to \mathcal{E}^{K'}$ and $\rho : \mathcal{R}^K \to \mathcal{R}^{K'}$ so that the product function $\eta \cdot \rho \cdot \eta$ maps any triple $r(h, t) \in \mathcal{T}^K$ to $\phi(r(h, t)) = \rho(r)(\eta(h), \eta(t)) \in \mathcal{T}^{K'}$. $\phi(K) = \big(\eta(\mathcal{E}^K), \rho(\mathcal{R}^K), \phi(\mathcal{T}^K)\big)$, the *image KG* of $K$ by $\phi$, is a subgraph of $K'$. A homomorphism $\phi$ is said to be *injective* if $\eta$ and $\rho$ are both injective. An injective KG homomorphism $\phi$ is called *KG monomorphism* and its mapping is represented by $\phi : K \xrightarrow{mon} K'$.

## 3.3 Graphlets and Motifs

The objective of our proposed method is to represent relational invariances and develop a structural vocabulary. To this end, the method identifies different graphlet occurrences within a KG. Usually graphlets are defined as induced subgraphs with respect to a Knowledge Graph while motifs are graphlets that occur frequently in a given KG (Pržulj, 2007; Milo et al., 2002; Ribeiro et al., 2021). To represent relational patterns in a more nuanced way, the following definitions are employed.

**Definition 3.1** (Graphlet). A graphlet $\mathcal{G} = (G, \mathrm{g})$ is a (small) connected Knowledge Graph $G = (\mathcal{E}, \mathcal{R}, \mathcal{T})$ with an order g on $\mathcal{R}$. $\mathrm{g}(\mathcal{R}) = \mathrm{g}(r_1, r_2, \ldots, r_m)$ a tuple defining the order on $\mathcal{R} \ni r_i$. The cardinality of a graphlet $\mathcal{G}$ is the number of its edges, $card(\mathcal{G}) = |\mathcal{T}|$.

The order specifies the manner in which directed relations within $\mathcal{R}$ connect their endpoints to create the graph structure. The order can relate two or many relations in $\mathcal{R}$; it is called *binary* and *n-ary* respectively. The arity ($n$) can be reduced from $n$ to $m$ with $m \leq n$ by only considering $m$ relations as arguments and the $n - m$ remaining relations become dummy arguments. This order is referred to as a *positional m-ary order*; it is denoted by $\mathrm{g}_{i_1, \ldots, i_m \leq n}$ when the attention is to stress on the position of the arguments and the initial arity. Thus $\mathrm{g}_{1,2,3 \leq 3}$ is an ordinary 3-ary order whereas $\mathrm{g}_{1,3 \leq 3}$ is a positional binary order.

**Theorem 3.2.** *Any positional m-ary order, $m < n$, spans a group of n-ary orders.*

In particular, the tuple $\mathrm{g}_{1,3 \leq 3}(\mathcal{R})$ could be induced by any of $\mathrm{g}_{1,1,3}(\mathcal{R}), \mathrm{g}_{1,3,3}(\mathcal{R})$ and $\mathrm{g}_{1,2,3}(\mathcal{R})$. In other words, if $\mathrm{g}_{1,3 \leq 3}(\mathcal{R})$ does not exist, then neither $\mathrm{g}_{1,1,3}(\mathcal{R}), \mathrm{g}_{1,3,3}(\mathcal{R})$ nor $\mathrm{g}_{1,2,3}(\mathcal{R})$ exist.

**Definition 3.3** (Graphlet occurrence). Let $K = (\mathcal{E}^K, \mathcal{R}^K, \mathcal{T}^K)$ be a KG, and g a positional m-aray order. A graphlet $\mathcal{G} = (G, \mathrm{g})$ *occurs* in $K$ if and only if $G$ is monomorphic to $K$, and the monomorphism $\phi_{\mathrm{g}} : G \xrightarrow{mon} K$ is an order-preserving mapping; that is, $\rho$ maps $\mathrm{g}(\mathcal{R})$ to $\mathrm{g} \circ \rho(\mathcal{R}) = \mathrm{g}(\rho(r_1), \rho(r_2), \ldots, \rho(r_m))$. The tuple $\mathrm{g}(\mathcal{R})$ induces the set of tuples $\mathrm{g}(\mathcal{R}^K) = \Big\{ \mathrm{g} \circ \rho(\mathcal{R}) | \rho : \mathcal{R} \xrightarrow{mon} \mathcal{R}^K \Big\}$. The image graph $\phi_{\mathrm{g}}(G)$, is called an *occurrence* of $\mathcal{G}$ in $K$.

The set of occurrences of $\mathcal{G}$ in $K$ is denoted and defined by $\mathcal{G}(K) = \Big\{ \phi_{\mathrm{g}}(G) | \phi_{\mathrm{g}} : G \xrightarrow{mon} K \Big\}$. Two occurring subgraphs $\phi(G)$ and $\phi'(G)$ are said *equivalent*, $\phi(G) \equiv_{\mathrm{g}} \phi'(G)$, if $\rho(r_{i_j}) = \rho'(r_{i_j})$ for $j \leq m, i_j \leq n$, and $r_{i_j} \in \mathcal{R}$. The equivalence class denoted by $\overline{\phi(G)} = \{\phi'(G) | \phi'(G) \equiv_{\mathrm{g}} \phi(G)\}$ is the set of subgraph occurrences equivalent to $\phi(G)$ and $\overline{\mathcal{G}(K)} = \Big\{ \overline{\phi(G)} | \phi(G) \in \mathcal{G}(K) \Big\}$ is the set of all equivalence classes.

The fact that two classes $\overline{\phi(G)}$ and $\overline{\phi'(G)}$ are different if and only if $\rho(\mathcal{R}) \neq \rho'(\mathcal{R})$ implies that there is a one-to-one correspondence between $\overline{\mathcal{G}(K)}$ and $\mathrm{g}(\mathcal{R}^K)$. We can therefore represent an equivalence class by the class $\overline{\phi(G)}$ or the tuple $\mathrm{g} \circ \rho(\mathcal{R})$.

The set of all graphlets of cardinality less than four is displayed in Figure 2. Graphlets play the role of the smallest structural entity in a Graph and are therefore well suited to investigate the local and global structure of a KG.

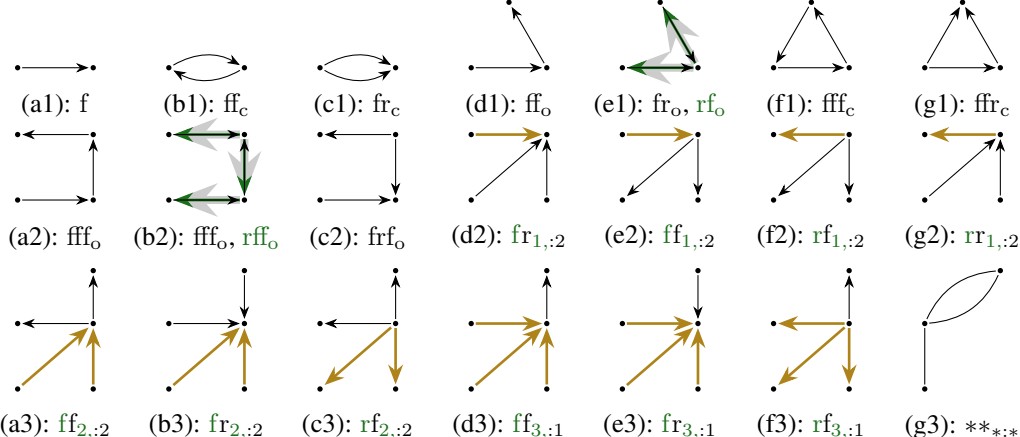

**Figure 2: Graphlets of size less than 5.** f and r denote forward and reverse edges, and subscripts c and o indicate closed and open paths. The green head arrows (shown with a light gray halo for clarity) form alternative graphlets, which are also indicated by the green labels to the right of the black text labels. The golden arrows, together with the black arrows, form distinct topological graphlets. Each vertex is marked with a small filled node for readability. The last four graphlets shown in the third column are not included in our approach.

## 4 METHOD

Our model $\text{Ultra}^+$ is a generalized extension of the Ultra framework introduced by Galkin et al., advancing its capabilities in relational pattern learning for KG reasoning. While the original Ultra relies solely on length-2 paths to define relational dependencies, $\text{Ultra}^+$ extends this approach by incorporating a richer set of graphlet-based patterns, capturing more complex and higher-order interactions between relations. In contrast to Motif, $\text{Ultra}^+$ constructs a binary relation graph using graphlets induced by positional binary orders, thereby preserving pairwise semantics. This shift allows $\text{Ultra}^+$ to encode cycles and subgraph patterns without resorting to hypergraph complexity.

### 4.1 A STRUCTURAL VOCABULARY FOR KNOWLEDGE GRAPH FOUNDATION MODELS

The structural vocabulary used to construct the relation graph constitutes the fundamental basis of $\text{Ultra}^+$.

**Definition 4.1.** A structural vocabulary over a KG, $K$, is a finite set $\mathcal{V} = \{(G_i, \text{g}_i), i \leq n_V\}$ of graphlets, and a weighting function

$$\omega : \bigcup_i \overline{\mathcal{G}_i(K)} \to \mathbb{N}, \quad \overline{\phi_{\text{g}}(G)} \mapsto |\overline{\phi_{\text{g}}(G)}| \tag{1}$$

mapping equivalence classes of occurrences to their cardinalities. The Knowledge Graph $K$ can be a union of KGs, that is $K = \cup_k (\mathcal{E}^k, \mathcal{R}^k, \mathcal{T}^k)$, and the domain of $\omega$ becomes $\bigcup_{i,k} \overline{\mathcal{G}_i(K_k)}$.

**Definition 4.2.** Let $K = (\mathcal{E}^K, \mathcal{R}^K, \mathcal{T}^K)$ be a KG, and $\mathcal{V}$ a structural vocabulary over $K$. We denote the *relation graph* over $K$ upon the structural vocabulary $\mathcal{V}$ by $\mathfrak{K} = (\mathfrak{E}, \mathfrak{R}, \mathfrak{T})$ where the set of nodes $\mathfrak{E} = \mathcal{R}^K$ and meta-relations $\mathfrak{R} = \mathcal{V}$.

A structural vocabulary of binary orders yields relation graphs; conversely, relation hypergraphs are generated. The (hyper)edges in $\mathfrak{T}$ are the tuples $\text{g}_i \circ \rho(\mathcal{R})$ . The tuple $\text{g} \circ \rho(\mathcal{R})$ does not exist unless its weight is nonzero. We say g is an $\varepsilon$-edge between the relations $\rho(r_1), \cdots, \rho(r_m) \in \mathcal{R}^K$, if and only if $\omega(\text{g} \circ \rho(\mathcal{R})) = \varepsilon$. To define the structural vocabulary, we did not specify any graphlet. This ensures that our framework can accommodate any type of graphlet. $\text{Ultra}^+$'s structural vocabulary is restricted to (short) path and topology based vocabularies.

**Path-Based Vocabulary.** In a graph, $K$, a path of length $p$, $\mathcal{P}_p = \{r_i(e_i, e_i'), i = 1, \cdots, p\}$ is a subset of $\mathcal{T}^K$ such that two consecutive triples share one entity. These paths are occurrences of the nine

graphlets shown in Figures 2(b1–c2). We shall note that $\mathrm{rr_o}(r_1, r_2) = \left\{ r_1(e_2, e_1), r_2(e_3, e_2) \in \mathcal{T}^K \right\}$ $= \mathrm{ff_o}(r_2, r_1)$. That is, $\mathrm{rr_o}$ can be substituted by $\mathrm{ff_o}$. In summary, the $\mathcal{P}_2$-based vocabulary $\mathcal{V}_2 = \{\mathrm{ff_{o,c}}, \mathrm{fr_{o,c}}, \mathrm{rf_o}\}$ is sufficient to characterize all closed or open 2-paths in $K$. In general, we define two distinct families of $\mathcal{P}_p$-based vocabularies $\mathcal{V}_p = \left\{ \mathrm{u} \smile \mathrm{v_z} | \mathrm{u}, \mathrm{v} \in \{\mathrm{f}, \mathrm{r}\}, \smile \in \{\mathrm{f}, \mathrm{r}\}^i, i = 0, \dots, p-2, \right.$ $\mathrm{z} \in \{\mathrm{o}, \mathrm{c}\} \}$ and $\mathcal{U}_p = \left\{ \mathrm{u} \smile \mathrm{v} | \mathrm{u}, \mathrm{v} \in \{\mathrm{f}, \mathrm{r}\}, \smile \in \{\mathrm{f}, \mathrm{r}\}^i, i = 0, \dots, p-2, \right\}$. $\smile$ is any sequence of length $i$ over the alphabets $\mathrm{f}, \mathrm{r}$. $\mathrm{u} \smile \mathrm{v_z}$ are positional binary orders relating the first and last relations appearing in a path of length $p > 1$. For $p = 2, \mathrm{u} \smile \mathrm{v_z} = \mathrm{uv_z}$ and for $p = 3$, $\mathrm{u} \smile \mathrm{w_z} :=$ $\mathrm{uvw_z} \in \left\{ \mathrm{fff_z}, \mathrm{ffr_z}, \mathrm{frf_z}, \mathrm{rff_z} | \mathrm{z} \in \{\mathrm{o}, \mathrm{c}\} \right\}$. It follows that $\mathcal{V}_m \subset \mathcal{V}_n$ if $m \le n$. This remains valid for the $\mathcal{U}_m$. It is imperative to note that the $\mathrm{u} \smile \mathrm{v}$ positional binary orders are incapable of discerning between closed and open paths. We designed variants of $\mathrm{Ultra}^+$ using these two families of structural vocabularies.

**Topology-based Vocabulary.** The degree of an entity in a KG is the sum of incoming and outgoing relations. The average number of degree per entity informs on the sparsity or density of the KGs. The type of relations surrounding an entity allow us to extract a subgraph centered on that entity, called an *m-star*, where $m$ is the degree of that entity. These m-stars are occurrences of graphlets that form the topology based vocabulary, denoted by $\mathcal{M}_m$. In m-stars, we count how many times each relation appears around the centered entity. For two relations, we have $i + j = m > 2$ and any of the relation can be an ingoing or outgoing relation. We write $\mathcal{M}_{ij}$, to emphasize on the degree of the relations. Figure 2 depicts $\mathcal{M}_{12} = \{uv_{1:2} \mid u, v \{f, r\}\} = \mathcal{M}_{21} = \mathcal{M}_3, \mathcal{M}_4' = \mathcal{M}_3 \cup \mathcal{M}_{22}$ and $\mathcal{M}_4 = \mathcal{M}_3 \cup \mathcal{M}_4'$. We combined the $\mathcal{V}_2$ and the $\mathcal{V}_3$ with the $\mathcal{M}_4'$ vocabularies to design two variants of $\mathrm{Ultra}^+$.

## 4.2 Representation Learning

KG representation learning consists of learning the entity and relation embeddings while preserving the KG structure. In our context, relations in $\mathcal{R}$ are entities in $\mathfrak{R}$. This duality leads to two representations, as described below.

**Relation Embedding.** $\mathrm{Ultra}^+$ embeds relations (nodes of the relation graph $\mathfrak{K}$) into $d_L$ dimensional real vectors, $\mathbf{h}_{r|q}^{(L)}$, by an L-layer message passing GNN, $GNN_{\mathcal{G}}$ . Following (Galkin et al., 2023), $\mathrm{Ultra}^+$ conditioned $(r|q)$ the embedding of relations $r$ on the query triple $q(h, ?)$. The input layer is initialized to $\mathbf{h}_{r|q}^{(0)} = \delta_{r=q} \mathbf{1}^{d_0}$, where $\delta_{r=q} = 1$ if $r = q$, and 0 otherwise. The following iterative process defines how the upcoming layers compute the embeddings

$$\mathbf{h}_{r|q}^{(t+1)} = \mathrm{UP}\big(\mathbf{h}_{r|q}^{(t)}, \mathrm{AGG}\big[\big\{\mathrm{MSG}\big(\big\{\big(\mathbf{h}_{r'|q}^{(t)}, \mathrm{u} \smile \mathbf{v_z}\big) | r' \in \mathcal{N}(\mathrm{u} \smile \mathrm{v_z}, \mathrm{r}), u \smile v_z \in \mathcal{V}\big)\big\}\big]\big)$$

so that $\mathbf{R}_{|q} = GNN_{\mathcal{G}}\left(\Theta_u, \Theta_a, \Theta_m, q, \mathfrak{R}\right) \in \mathbb{R}^{|\mathcal{R}| \times d_L}$ is the conditional relation embedding matrix of all relations in $\mathfrak{E}$. UP, AGG and MSG are *update, aggregation, and message passing functions* and $\Theta_x, x = u, a, m$ are their respective parameters.

**Entity Embedding.** Entities are first initialized conditionally to the query $q(h, ?)$ and the relation embedding $\mathbf{q}$, a vector column of $\mathbf{R}_{|q}$. We iteratively embed entities as follows

$$\mathbf{h}_{e|h,q}^{(0)} = \delta_{e=h} \mathbf{q}$$
$$\mathbf{h}_{e|h,q}^{(t+1)} = \mathrm{UP}\big(\mathbf{h}_{e|h,q}^{(t)}, \mathrm{AGG}\big[\big\{\mathrm{MSG}\big(\big\{\big(\mathbf{h}_{e'|h,q}^{(t)}, \mathrm{f}^t(\mathbf{q})\big) | e' \in \mathcal{N}(r, e), r \in \mathcal{R}\big)\big\}\big]\big)$$
$$\pi(h, q, e) = \mathbf{w}^\top\big(\mathbf{W}^{L'} \mathbf{h}_{e|h,q}^{(L')} + \mathbf{b}^{L'}\big) + b.$$

Motivated by the ability of geometric KGEs to capture complex relational patterns through algebraic transformations, the message-passing function is enriched with non-linear layer-specific relational transformations $\mathrm{f}^t$. The transformations, $\mathrm{f}^t(\mathbf{q}) = \mathbf{W}_2^t \mathrm{ReLU}\left(\mathbf{W}_1^t \mathbf{q} + \mathbf{b}_1^t\right) + \mathbf{b}_2^t$ are 2-layer perceptrons with the ReLU activation function; $\mathbf{W}$ are matrices, $\mathbf{b}$ and $\mathbf{w}$ are vectors, and $b$ is a scalar. The update functions consist of a linear transformation followed by a normalization layer, while aggregation is performed through summation. $\pi(h, q, e)$ is the score of the triple $q(h, e)$. The initialization of relations to vectors of ones, $\mathbf{1}^{d_L}$, or zeros, $\mathbf{0}^{d_L}$, and entities to $\mathbf{q}$ or zero vectors, makes the architecture of $\mathrm{Ultra}^+$ generalizable to unseen relations and entities during inference. $\mathrm{Ultra}^+$ uses

the binary cross entropy (BCE) loss,

$$\mathcal{L}_{\text{BCE}} = -\frac{1}{|\mathcal{T}_+|} \sum_{\tau \in \mathcal{T}_+} \Big( \log \pi(\tau) + \frac{1}{n(\tau)} \sum_{i=1}^{n(\tau)} \log \big( 1 - \pi(\tau_i') \big) \Big)$$

to measure the difference between predicted probabilities and triple plausibilities. The BCE loss penalizes high score for true triples, $\tau$, low score for corrupted triples, $\tau_i'$.

### 4.3 COMPARING Ultra$^+$ AND MOTIF

The KGFM Motif uses a variety of motifs (n-ary orders) to construct a relation hypergraph. The experiments in (Huang et al., 2025) are conducted on 2-path, 3-path, and k-star motifs, denoted by $\mathcal{F}_k^{path}$ and $\mathcal{F}_k^{star}$, respectively. It can be observed that Motif is unable to discriminate between closed and open paths, unlike Ultra$^+$. The second significant distinction derived from the orders. To explain this, let us consider the motifs arising from their respective 3-path based structural vocabulary. The 3-aries in Motif are named and are equivalent to ours as follows: tfh $\sim$ fff, tft $\sim$ ffr, hfh $\sim$ frf and hft $\sim$ rff. As the IKG in Figure 3a is a directed acyclic graph, the u$\smile$v are equivalent to u$\smile$v$_o$. Figure 3b is therefore built using the latter. From Figure 3a, $r_1, r_2$ and $r_3$ are linked by the motif tfh and $(a, r_1, b, r_2, c, r_3, d)$ is the only element in the equivalence class tfh$(r_1, r_2, r_3)$. Similarly tfh$(r_1, r_4, r_3)$ and tfh$(r_1, r_5, r_3)$ are singletons. However, the equivalence class fff$_o(r_1, r_3)$ is the union of tfh$(r_1, r_i, r_3), i = 2, 4, 5$, this is to say $\omega\left(\text{fff}_o\left(r_1, r_2, r_3\right)\right) = \sum_i \omega\left(\text{tfh}\left(r_1, r_i, r_3\right)\right)$. In general, the weights of the equivalence classes induced by Ultra$^+$'s 3-path motifs are higher than the Motif's ones.

**Theorem 4.3.** *Let $\rho$ be a monomorphism from $\mathcal{P}_3$ to a graph $K$. If* uvw$_o \circ \rho$ *is an $\epsilon$-edge and its corresponding motif in* Motif*'s vocabulary is an $\epsilon'$-edge, then $\epsilon' \leq \epsilon$.*

Theorem 4.3 states that if no edge exists between two relations in the Ultra$^+$ relation graph, then they are not connected by a hyper-edge in the Motif relation hypergraph. This proves the robustness of Ultra$^+$. Furthermore, the theorem demonstrates that Ultra$^+$ is computationally less demanding than Motif. This difference in computation appears in the $\text{GNN}_\mathcal{G}$'s message function. In order to clarify this statement, let us consider the neighborhoods of $r_3$ in both relation graphs. Ultra$^+$ returns $\mathcal{N}(\text{fff}_o, r_3) = \{r_1\}$ while Motif returns $\mathcal{N}^1(\text{tfh}, r_3) = \emptyset$, $\mathcal{N}^2(\text{tfh}, r_3) = \emptyset$ and $\mathcal{N}^3(\text{tfh}, r_3) = \{r_1, r_2, r_4, r_5\}$; where the upper script $^i$ means $r_3$ appears at the $i$'th position in the hyperedge tfh. In comparison to $\mathcal{N}(\text{fff}_o, r_3)$, operations over $\mathcal{N}^3(\text{tfh}, r_3)$ result in an increase in compute time. The choice of relation embedding GNN also contributes to the increases of computing time. The HCNets used by Motif genuinely involves more computation, as it employs a learnable query vector and a sinusoidal positional encoding for each query relation $q$.

## 5 EXPERIMENTS AND RESULTS

In our experiments, we aim to answer the following research questions:

**(RQ1)** Can the scaling behavior of recent GNN based graph foundation models be improved with the proposed extension?

**(RQ2)** Does the zero shot performance increase with increasing vocabulary?

**(RQ3)** Can the addition of specific topological graphlets (e.g., N-M graphlets) help link prediction for containing N-M relations?

**(RQ4)** Does enriching vocabulary with closed paths improve model performance?

**(RQ5)** Does constructing a relation graph with binary meta-relations offer advantages over using ternary meta-relations?

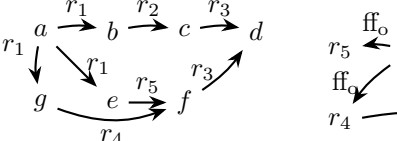

(a) Illustrative Toy KG (IKG)

(b) Relation graph of the IKG

Figure 3: (a) A toy Knowledge Graph (IKG) with five relations and seven entities, illustrating the underlying relational structure. (b) The corresponding relation graph constructed from the structural vocabulary of open paths $\{\text{ff}_o, \text{fff}_o\}$, where relations are nodes and edges capture their co-occurrence within paths.

**Benchmarks and Pattern-Matched Relation Graphs.** In our experiments, we assess the essence of Ultra$^+$ using 57 KGs with various characteristics. We grouped these KGs according to entity

learning (18 KGS), graph transfer (23 KGs), and transductive learning (16 KGs) tasks. Additional information about the KGs in each group is included in Appendix B. Ultra$^+$ is pretrained on the CoDEx Medium, FB15k237, and WN18RR KGs, and subsequently assessed in the ZSLP tasks.

Galkin et al. (2023) and Huang et al. (2025) employ sparse matrix multiplication of the (multi-relational) adjacency matrix $\mathbb{A} \in \mathbb{R}^{n \times m \times n}$ and matrices representing head-relation pair $\mathbb{E}_h \in \mathbb{R}^{n \times m}$ and tail-relation pair $\mathbb{E}_t \in \mathbb{R}^{n \times m}$. Multiplying $\mathbb{E}_x^T$ by $\mathbb{E}_y$ results into the adjacency matrix of the x2y meta-relations used in Ultra. This method provides additional information on the number of occurrences of the respective graphlet in the whole dataset. However, this information is left unused, as only the connection information is represented in the relation KG in Ultra and the relation hypergraph in Motif, respectively. As we also distinguish between closed and open path, computation via the adjacency matrix becomes computationally expensive. Pattern matching, on the other hand, can be used in a highly parallel fashion to compute the relation graph of any KG. To obtain the relation graph we construct a SPARQL ask query for each element in the vocabulary (see Appendix E.4 for the exact Queries), which can be run on any rdf KG. This method enables the computation of relation graphs based on vocabularies containing arbitrary graphlets.

**Evaluation Protocol.** We follow the best practices for evaluating KGE models by considering the Mean Reciprocal Rank (MRR), and the Hits at n (Hn, n = 1,3,10) metrics. Link prediction consists of finding the missing entity ? in the queries $Q = r(h, ?)$ or $Q' = r(?, t)$. First, we created a symbolic inverse relation $r'$, which turns queries with a missing head into $Q = r'(t, ?)$. This means that we only look at queries that are in the form of $Q$. Next, Ultra$^+$ scores and ranks the corrupted triples in a decreasing order. The predicted missing entity is the top ranked corrupted entity. We compare our models against the state-of-the-art KGFMs Ultra and Motif using the aforementioned evaluation metrics. We consider six different vocabularies to design our models; namely, $\mathcal{U}_2, \mathcal{V}_j^- = \mathcal{V}_j \setminus \{u \smile v_c\}, \mathcal{V}_j, \mathcal{V}_j^+ = \mathcal{V}_j \cup \mathcal{M}_4', j = 2, 3$ . We denote the Ultra$^+$ variant built on the vocabulary $\mathcal{X}_\bullet^\pm$ by Ultra$^+[\mathcal{X}_\bullet^\pm]$.

**Results.** The experimental results of evaluating the pretrained Ultra$^+$ on the benchmark KGs are reported in Table 1. In the following, the operator $\geq$ relating two models means that the first model *outperforms* the second model.

Table 1: Average zero-shot link prediction MRR and H10 over 51 KGs. Baseline results are taken from (Huang et al., 2025). $\mathcal{P}_n$, O and C stand for n-, open, and closed paths; and N-M stands for many-to-many subgraphs

| Model | Structural Vocabulary | | | Ind.$(e)$ (18 KGs) | | Ind.$(e, r)$ (23 KGs) | | Transd. (10 KGs) | | Total Avg. (51 KGs) | |
|---|---|---|---|---|---|---|---|---|---|---|---|
| | $\mathcal{V}$ | Definition | $\#\mathcal{V}$ | MRR | H10 | MRR | H10 | MRR | H10 | MRR | H10 |
| Ultra$^+$ | $\mathcal{V}_2^-$ | O. $\mathcal{P}_2$ | 4 | .388 | .551 | .323 | .498 | .338 | .498 | .349 | .516 |
| | $\mathcal{U}_2$ | $\mathcal{P}_2$ | 4 | .425 | .567 | .350 | .515 | .343 | .499 | .375 | .530 |
| | $\mathcal{V}_2$ | O. & C. $\mathcal{P}_2$ | 8 | .441 | .579 | .354 | .533 | .349 | .509 | .384 | .544 |
| | $\mathcal{V}_2^+$ | O. & C. $\mathcal{P}_2$ & N-M | 16 | .415 | **.582** | .349 | .525 | .347 | .504 | .372 | .541 |
| | $\mathcal{V}_3^-$ | O. $\mathcal{P}_3$ | 16 | .423 | .561 | .337 | .510 | .346 | .496 | .369 | .525 |
| | $\mathcal{V}_3$ | O. & C. $\mathcal{P}_3$ | 24 | **.445** | .581 | .355 | **.542** | **.355** | **.511** | **.387** | **.549** |
| | $\mathcal{V}_3^+$ | O. & C. $\mathcal{P}_3$ & N-M | 32 | .435 | .581 | **.356** | .532 | .345 | .499 | .382 | .543 |
| Ultra | $\mathcal{U}_2$ | $\mathcal{P}_2$ | 4 | .431 | .566 | .345 | .513 | .339 | .494 | .374 | .529 |
| Motif | $\mathcal{U}_3$ | $\mathcal{P}_3$ | 12 | .436 | .577 | .349 | .525 | .343 | .496 | .378 | .537 |

*General Overview:* Figure 4 reports the average performance of Ultra and Ultra$^+$ as the number of pretraining KGs increases. (**RQ1**) Ultra$^+$'s variants consistently outperform Ultra, demonstrating both scalability and performance gains. While Ultra$^+[\mathcal{V}_2]$ improves monotonically before saturating, Ultra$^+[\mathcal{V}_3]$ fluctuates with the addition of WN18RR and ConceptNet100k at position 2 and 6 respectively (see Table 16 for more details), highlighting the need for careful selection and ordering of pretraining KGs given their structural heterogeneity. Observing the path- and topology- based vocabulary variants, we notice that Ultra$^+[\mathcal{V}_3] \geq$ Ultra$^+[\mathcal{V}_2] \geq$ Ultra$^+[\mathcal{U}_2] \geq$ Ultra$^+[\mathcal{V}_2^-]$ on average for relation learning, graph transfer, and transductive inference tasks. This trend in performance can be related to the inclusion of the vocabularies $\mathcal{V}_2^- \subset \mathcal{U}_2 \subset \mathcal{V}_2 \subset \mathcal{V}_3$. However, we have found

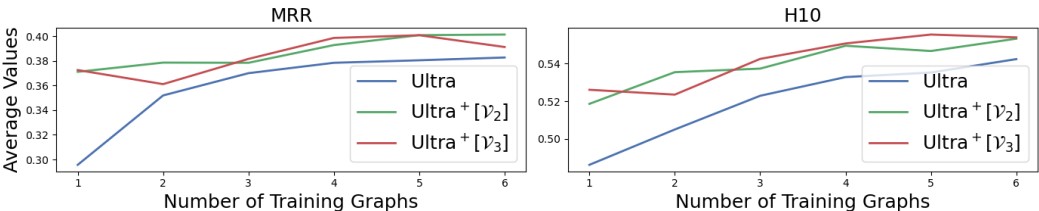

Figure 4: Average Performance over 51 Graphs of Ultra and Ultra$^+$ models pretrained on an increasing number of Graphs.

that combining the path-based and topology-based vocabularies does not result in an increase in performance as Ultra$^+[\mathcal{V}_3]$ and Ultra$^+[\mathcal{V}_2]$ often surpass Ultra$^+[\mathcal{V}_3^+]$ and Ultra$^+[\mathcal{V}_2^+]$ respectively. **(RQ2)** On one hand, we can conclude that the zero-shot performance increases with increasing the path-based vocabulary. **(RQ3)** On the other hand, mixing the topology-based with the path-based vocabulary does not necessarily preserve the performance increase.

The Motif model maintains its superiority over Ultra for all tasks. This difference in performance is a consequence of adding 3-path graphlets to Ultra's vocabulary. Although Ultra$^+[\mathcal{V}_2]$ utilized only the 2-path based vocabulary $\mathcal{V}_2$, it notably outperforms Motif and Ultra on both inductive and transductive link prediction. **(RQ4)** This clearly demonstrates the importance of having a vocabulary rich enough to convey information about closed and open paths. Ultra$^+[\mathcal{V}_3]$ is the best performing variant of the Ultra$^+$ models across all the settings. It uses the same vocabulary as Motif, except that its graphlets are positional binary orders. **(RQ5)** Thus, its superiority over Motif lies in the arity of the graphlets.

***On the robustness of* Ultra$^+$:** In all three models: Ultra$^+$, Ultra, and Motif, two relations are connected in the relation graph as soon as they co-occur at least once in the KG. For Ultra$^+$, this corresponds to observing a single match of the associated SPARQL query pattern in the KG; for Ultra and Motif, it corresponds to the relevant entry of the adjacency matrix (obtained via sparse matrix multiplication) becoming non-zero. In either case, the frequency of co-occurrence does not influence whether an edge is created. However, because Ultra$^+$ employs positional binary orders, this insensitivity to frequency is implicitly mitigated in its relation graph, as formalized in Theorem 4.3. Empirically, sparse KGs provide a natural setting for assessing the robustness of KGFMs that construct relation graphs. In our experiments, WDsinger, NELL23k, and FB15k237(10/20/50) constitute such sparse knowledge graphs.

Table 2: Comparing Ultra, Motif and Ultra$^+$ on 5 transductive sparse datasets. Baseline results are taken from (Huang et al., 2025).

| Model | Ultra | | Motif | | Ultra$^+[\mathcal{V}_2]$ | | Ultra$^+[\mathcal{V}_3]$ | |
|---|---|---|---|---|---|---|---|---|
| **Dataset** | **MRR** | **H10** | **MRR** | **H10** | **MRR** | **H10** | **MRR** | **H10** |
| WDsinger | .382 | .498 | .397 | **.514** | **.402** | .505 | **.402** | .511 |
| NELL23k | .239 | .408 | .220 | .384 | .249 | .413 | **.250** | **.419** |
| FB15k237(10) | .248 | .398 | .236 | .384 | **.249** | **.404** | .245 | .400 |
| FB15k237(20) | .272 | .436 | .259 | .422 | **.274** | **.439** | .268 | .431 |
| FB15k237(50) | .324 | .526 | .312 | .508 | **.329** | **.527** | .326 | .524 |

## 6 CONCLUSION

We proposed a KGFM framework called Ultra$^+$ capable of constructing a relation graph from any structural vocabulary composed of a set of graphlets. This framework enables the conversion of n-ary graphlets' orders into positional binary orders, thereby maintaining pairwise relational semantics and mitigating the complexity associated with hypergraphs. Using SPARQL to run ASK queries simplifies the distinction between open and closed paths when mining graphlets, and overcomes the major limitation of computing relation graphs when higher-order graphlets involve the full adjacency matrix. Our theoretical findings, described in Theorems 3.2 and 4.3, demonstrate that Ultra$^+$

exhibits greater robustness compared to the current baseline KGFMs. Evaluation of ZSLP tasks, with $\text{Ultra}^+$ pretrained on three KGs, indicated that an increase in structural vocabulary is advantageous when only path-based vocabulary is utilized, yet it becomes detrimental when combining path- and topology-based vocabularies. We showed that enhancing the model's awareness of relational patterns and topological patterns significantly improves the model's MRR and H10, respectively.

Our Model $\text{Ultra}^+[\mathcal{V}_3]$ achieves state-of-the-art performance in ZSLP averaged across 51 datasets with only 3 Graphs used for pretraining. Our investigation also shows that scaling pretraining has the chance to further improve performance. The case for scaling the vocabulary, on the other hand, remains ambiguous. We observed an increase in performance for increasing path based vocabulary, while adding structurally inspired graphlets seems to be detrimental. A large scale investigation of larger structural vocabularies remains challenging, due to the computational complexity of Relation Graph computation for vocabularies containing complex graphlets. We will address efficient computation of Relation Graphs that go beyond instance based computation (where existence of a single instance of a grpahlet results in a connection in the Relation Graph) in future research.

## REPRODUCIBILITY STATEMENT

We publish our full implementation in an open repository: ultra-augmentations[1]. It contains model architectures, training scripts, hyperparameter configurations, dataset preprocessing code, and random seeds as well as trained checkpoints. Detailed instructions for reproducing all experiments are included to guarantee full reproducibility of our results.

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

# A    PROOF OF THE THEOREMS

In this section, we provide the proofs for the theorems stated in the paper. Before we begin, let us summarize the key mathematical symbols and their meanings used throughout the paper in Table 3

## A.1    PROOF OF THEOREM 3.2

Any positional m-ary order, $m < n$, spans a group of n-ary orders.

*Proof.* Consider two positive integers, $m$ and $n$, where $m < n$. Let $g = g_{i_1,\ldots,i_m \leq n}$ represents a positional m-ary. According to Definition 3.1, g is a function with $n$ arguments, out of which $n - m$ are dummy arguments. Define $\pi(i_k)$ as the position of $i_k$ within the ordered list of all $n$ arguments. We now consider the family of n-ary $g^{(j)} = g_{j_1,\ldots,j_n}$ such that if $\pi(j_l) = \pi(i_k)$, then $j_l = i_k$ for $k = 1,\ldots,m$ and $l = 1,\ldots,n$. Consequently, g is induced by any of the n-ary $g^{(j)}$. In other words, the m-ary spans the aforementioned family of n-aries. $\qquad\square$

Table 3: Notation Table.

| Symbol | Meaning |
|---|---|
| $\mathcal{E}, \mathcal{R}$ | Entity and relation sets |
| $\mathcal{T}, +, -$ | Set of all plausible, true $(+)$, and corrupted $(-)$ triples |
| $K = (\mathcal{E}^K, \mathcal{R}^K, \mathcal{T}^K_+)$ | Knowledge Graph |
| $\phi, \rho, \eta$ | KG, relation, and entity homomorphism |
| $\phi_{\mathrm{g}} = (\rho, \eta) : G \xrightarrow{mon} K$ | KG monomorphism with $\rho : \mathcal{R} \to \mathcal{R}^K, \eta : \mathcal{E} \to \mathcal{E}^K$ two injective maps |
| $card(\mathcal{G}) = |\mathcal{T}|$ | Cardinality of the graphlet $\mathcal{G}$ |
| $\mathcal{G}, G, \mathrm{g}$ | Graphlet, small KG, and order on $\mathcal{R}$ |
| $\mathrm{g}(\mathcal{R})$ | A tuple defined by the order g on $\mathcal{R}$ |
| $\mathrm{g}_{i_1,\ldots,i_{m \leq n}}$ | (if $m < n$) positional m-, (else) n-ary order |
| $\mathrm{g}(\mathcal{R}^K) = \left\{ \mathrm{g} \circ \rho(\mathcal{R}) \mid \rho : \mathcal{R} \xrightarrow{mon} \mathcal{R}^K \right\}$ | Ordered n-aries (of relations in $\mathcal{R}^K$) induced by g |
| $\phi_{\mathrm{g}}(G), \mathcal{G}(K)$ | Occurrence, set of occurrences of $\mathcal{G}$ in $K$ |
| $\overline{X} = \{\phi'(G) \mid \phi'(X) \equiv_{\mathrm{g}} \phi(X)\}$ | Equivalence class |
| $(\mathcal{V}, \omega)$ | Structural vocabulary, $\mathcal{V} = (\mathcal{G}_i, \mathrm{g}_i)$ a set of graphlets, $\omega(\overline{\phi(G_i)})$ a weighting function |
| $\mathrm{f}, \mathrm{r}, {}_{\mathrm{o}}, {}_{\mathrm{c}}$ | Forward, backward/reversed, open, and closed path |
| $\smile \in \{\mathrm{f}, \mathrm{r}\}^i$ | A sequence of length $i$ over the alphabets $\mathrm{f}, \mathrm{r}$. |
| $u \smile v = \mathrm{g}_{1,n}$ | A Positional 2-ary defined by the first and last position |
| $\mathcal{P}_p$ | Set of paths of length $p$ |
| O., C., O. & C. $\mathcal{P}_p$ | Open, closed, and open or closed paths of length $p$. |
| $\mathcal{V}_p = \left\{ u \smile v_{\mathrm{z}} \mid u, v \in \{\mathrm{f}, \mathrm{r}\}, \mathrm{z} \in \{\mathrm{o}, \mathrm{c}\} \right\}$ | (O., C.) $\mathcal{P}_p$-based vocabulary |
| $\mathcal{U}_p = \left\{ u \smile v \mid u, v \in \{\mathrm{f}, \mathrm{r}\}, \right\}$ | $\mathcal{P}_p$-based vocabulary |
| N-M | Many to many |
| $\mathcal{M}_{ij} = \{uv_{i:j} \mid u, v \in \{\mathrm{f}, \mathrm{r}\}\}$ | N-M subgraphs with $i, j$ number of u and v edges resp. |
| $\mathcal{M}_m = \{\mathcal{M}_{i,j} \mid i + j = m\}$ | m-star |
| $\mathcal{V}_p^-$ | O. $\mathcal{P}_p$-based vocabulary |
| $\mathcal{V}_p^+$ | O. & C. $\mathcal{P}_p$- and $\mathcal{M}_4$-based vocabulary |
| $\mathcal{N}^i(\mathrm{g}, r)$ | Neighborhood of $r$ (the $i$th node) relative to the n-ary g |
| $\mathrm{Ultra}^+[\mathcal{X}_p^{\pm}]$ | $\mathrm{Ultra}^+$ variant built on the vocabulary $\mathcal{X}_p^{\pm}$ |

## A.2 PROOF OF THEOREM 4.3

Let $\rho$ be a monomorphism from $\mathcal{P}_3$ to a graph $K$. If $uvw_{\mathrm{o}} \circ \rho$ is an $\epsilon$-edge and its corresponding motif in $\mathrm{Motif}$'s vocabulary is an $\epsilon'$-edge, then $\epsilon' \leq \epsilon$.

*Proof.* Let $\rho$ be a monomorphism from $\mathcal{P}_3$ to a graph $K$. $uvw_{\mathrm{o}}$ is a positional binary order whose second argument is the dummy argument; i.e. $uvw_{\mathrm{o}} = \mathrm{g}_{1,3}$. It follows from Theorem 3.2 that $uvw_{\mathrm{o}}$ spans a group of 3-ary, $\mathrm{g}^{(j)}$, including it corresponding motif $\mu$. All triples in the equivalence classes $\mathrm{g}^{(j)} \circ \rho(\mathcal{R})$ are also in $uvw_{\mathrm{o}} \circ \rho(\mathcal{R})$, so that $\epsilon = \omega(uvw_{\mathrm{o}} \circ \rho(\mathcal{R})) = \sum_j \omega(\mathrm{g}^{(j)} \circ \rho(\mathcal{R})) \geq \mu \circ \rho(\mathcal{R}) = \epsilon'$ $\square$

## A.3 EXPRESSIVENESS LIMITATION OF MOTIF AUGMENTED WITH A CLOSED PATH COMPARE TO ULTRA$^+$

One of $\mathrm{Ultra}^+$'s contributions is distinguishing between closed and open paths. Although the GNN architectures of $\mathrm{Ultra}^+$ and $\mathrm{Motif}$ are different, we are interested in finding out whether $\mathrm{Motif}$ augmented with closed paths is more expressive than $\mathrm{Ultra}^+$. Let us assume that $\mathrm{Motif}$ is more expressive than $\mathrm{Ultra}^+$. We will now consider the cyclic KG in Figure 5 and its relation graph in the $\mathrm{Ultra}^+$ and $\mathrm{Motif}$ framework. As this graph is symmetric, the results are independent of the choice of query relation. Let $r_1$ be the query relation.

(a) Cyclic KG  (b) Relation (Binary)graph  (c) Relation Hypergraph

Figure 5: Cyclic Knowledge Graph and Relation Graphs: (a) A cyclic knowledge graph with three relations. (b) Ultra$^+$ constructs a relation graph consisting of three 2-ary edges, while (c) Motif constructs a relation hypergraph with a single 3-ary edge.

**Relation Encoding with Motif.** The relation graph, $G_M$, constructed in Motif framework has three edges, mainly: $\text{tfh}(r_1, r_2, r_3)$, $\text{tfh}(r_3, r_1, r_2)$, and $\text{tfh}(r_2, r_3, r_1)$. We have $\mathbf{h}^{(0)}_{r_1|r_1} = \mathbf{1}$ and $\mathbf{h}^{(0)}_{r_i|r_1} = \mathbf{0}, i \neq 1$. Thus,

$$\mathbf{h}^{(1)}_{r_1|r_1} = \text{UP}\big(\mathbf{h}^{(0)}_{r_1|r_1}, \text{AGG}\big[\big\{\text{MSG}\big(\big\{\big(\mathbf{h}^{(0)}_{r'|r_1}, \mathbf{tfh}\big)|r' \in \{r_2, r_3\}\big\}\big)\big\}\big]\big)$$
$$= \mathbf{1};$$
$$\mathbf{h}^{(1)}_{r_2|r_1} = \text{UP}\big(\mathbf{h}^{(0)}_{r_2|r_1}, \text{AGG}\big[\big\{\text{MSG}\big(\big\{\big(\mathbf{h}^{(0)}_{r'|r_1}, \mathbf{tfh}\big)|r' \in \{r_1, r_3\}\big\}\big)\big\}\big]\big)$$
$$= \mathbf{1} \times \mathbf{tfh} = \mathbf{tfh};$$
$$\mathbf{h}^{(1)}_{r_3|r_1} = \text{UP}\big(\mathbf{h}^{(0)}_{r_3|r_1}, \text{AGG}\big[\big\{\text{MSG}\big(\big\{\big(\mathbf{h}^{(0)}_{r'|r_1}, \mathbf{tfh}\big)|r' \in \{r_1, r_2\}\big\}\big)\big\}\big]\big)$$
$$= \mathbf{1} \times \mathbf{tfh} = \mathbf{tfh};$$

We observe that $\mathbf{h}^{(1)}_{r_2|r_1} = \mathbf{h}^{(1)}_{r_3|r_1}$ and assume that this holds for all $t \leq T$ where $T > 1$. It follows that

$$\mathbf{h}^{(T+1)}_{r_2|r_1} = \text{UP}\big(\mathbf{h}^{(T)}_{r_2|r_1}, \text{AGG}\big[\big\{\text{MSG}\big(\big\{\big(\mathbf{h}^{(T)}_{r'|r_1}, \mathbf{tfh}\big)|r' \in \{r_1, r_3\}\big\}\big)\big\}\big]\big)$$
$$= \mathbf{h}^{(T)}_{r_2|r_1} + (\mathbf{h}^{(T)}_{r_1|r_1} + \mathbf{h}^{(T)}_{r_3|r_1}) \times \mathbf{tfh};$$
$$\mathbf{h}^{(T+1)}_{r_3|r_1} = \text{UP}\big(\mathbf{h}^{(T)}_{r_3|r_1}, \text{AGG}\big[\big\{\text{MSG}\big(\big\{\big(\mathbf{h}^{(T)}_{r'|r_1}, \mathbf{tfh}\big)|r' \in \{r_1, r_2\}\big\}\big)\big\}\big]\big)$$
$$= \mathbf{h}^{(T)}_{r_3|r_1} + (\mathbf{h}^{(T)}_{r_1|r_1} + \mathbf{h}^{(T)}_{r_2|r_1}) \times \mathbf{tfh}$$
$$= \mathbf{h}^{(T)}_{r_2|r_1} + (\mathbf{h}^{(T)}_{r_1|r_1} + \mathbf{h}^{(T)}_{r_3|r_1}) \times \mathbf{tfh}$$
$$= \mathbf{h}^{(T+1)}_{r_2|r_1}.$$

Thus, Motif can't differentiate $r_2$ from $r_3$.

**Relation Encoding with** Ultra$^+$**.** On the other hand, from $\mathbf{h}^{(0)}_{r_1|r_1} = \mathbf{1}$ and $\mathbf{h}^{(0)}_{r_i|r_1} = \mathbf{0}, i \neq 1$, Ultra$^+$'s relation embedding yields

$$\mathbf{h}^{(1)}_{r_1|r_1} = \text{UP}\big(\mathbf{h}^{(0)}_{r_1|r_1}, \text{AGG}\big[\big\{\text{MSG}\big(\big\{\big(\mathbf{h}^{(0)}_{r'|r_1}, \mathbf{fff_c}\big)|r' \in \{r_2\}\big\}\big)\big\}\big]\big)$$
$$= \mathbf{1};$$
$$\mathbf{h}^{(1)}_{r_2|r_1} = \text{UP}\big(\mathbf{h}^{(0)}_{r_2|r_1}, \text{AGG}\big[\big\{\text{MSG}\big(\big\{\big(\mathbf{h}^{(0)}_{r'|r_1}, \mathbf{fff_c}\big)|r' \in \{r_3\}\big\}\big)\big\}\big]\big)$$
$$= \mathbf{0};$$
$$\mathbf{h}^{(1)}_{r_3|r_1} = \text{UP}\big(\mathbf{h}^{(0)}_{r_3|r_1}, \text{AGG}\big[\big\{\text{MSG}\big(\big\{\big(\mathbf{h}^{(0)}_{r'|r_1}, \mathbf{fff_c}\big)|r' \in \{r_1\}\big\}\big)\big\}\big]\big)$$
$$= \mathbf{1} \times \mathbf{fff_c} = \mathbf{fff_c}.$$

Since $\mathbf{h}_{r_2|r_1}^{(1)} \neq \mathbf{h}_{r_3|r_1}^{(1)}$, let us assume that $\mathbf{h}_{r_{2,3}|r_1}^{(t)}$ are distinct vectors for $t < T$ except for $\mathbf{h}_{r_2|r_1}^{(T)} = \mathbf{h}_{r_3|r_1}^{(T)} = \mathbf{c}_T$. We would then have

$$\mathbf{h}_{r_1|r_1}^{(T+1)} = \mathrm{UP}\big(\mathbf{h}_{r_1|r_1}^{(T)}, \mathrm{AGG}\big[\{\mathrm{MSG}(\{(\mathbf{h}_{r'|r_1}^{(T)}, \mathbf{fff_c})|r' \in \{r_2\}\})\}\big]\big)$$
$$= \mathbf{h}_{r_1|r_1}^{(T)} + \mathbf{c}_T \times \mathbf{fff_c};$$
$$\mathbf{h}_{r_2|r_1}^{(T+1)} = \mathrm{UP}\big(\mathbf{h}_{r_2|r_1}^{(T)}, \mathrm{AGG}\big[\{\mathrm{MSG}(\{(\mathbf{h}_{r'|r_1}^{(T)}, \mathbf{fff_c})|r' \in \{r_3\}\})\}\big]\big)$$
$$= \mathbf{c}_T + \mathbf{c}_T \times \mathbf{fff_c};$$
$$\mathbf{h}_{r_3|r_1}^{(T+1)} = \mathrm{UP}\big(\mathbf{h}_{r_3|r_1}^{(T)}, \mathrm{AGG}\big[\{\mathrm{MSG}(\{(\mathbf{h}_{r'|r_1}^{(T)}, \mathbf{fff_c})|r' \in \{r_1\}\})\}\big]\big)$$
$$= \mathbf{c}_T + \mathbf{h}_{r_1|r_1}^{(T)} \times \mathbf{fff_c}$$
$$\neq \mathbf{h}_{r_2|r_1}^{(T+1)}$$

since $\mathbf{h}_{r_1|r_1}^{(t)}$ and $\mathbf{c}_t$ are multivalued polynomials of indeterminate $\mathbf{fff_c}$ and constant terms $\mathbf{1}$ and $\mathbf{0}$ respectively. In other words, $\mathrm{Ultra}^+$ is able to distinguish between $r_2$ and $r_3$. This contradicts our assumption about the expressive power of Motif.

We can then conclude that $\mathrm{Ultra}^+$ is at least as expressive as Motif.

## B   DATASETS

Our Experiments have been performed on a multitude of datasets, following (Galkin et al., 2023). These datasets can be grouped into the following three subsets:

- **Inductive** $(e, r)$: Inductive link prediction datasets with prediction on new nodes and new relations.

- **Inductive** $(e)$: Datasets for inductive Link prediction on new nodes.

- **Transductive**: Transductive Link prediction on seen nodes and relations.

Datasets and corresponding statistics are displayed in tables 4-6.

## C   ADDITIONAL RESULTS

In tables 7 - 9 we display the zero shot results of Ultra and our models $\mathrm{Ultra}^+$ on all datasets. Here we show the models that have been pre-trained on a mixture of 3 graphs (see Table 16).

Additionally we performed finetuning of our best model on all datasets considered here. The detailed results are displayed in Tables 11, 10 and 12. For finetuning we employed dataset specific hyperparameters as displayed in table 14. Hyperparameters common to all datasets are in table 15. Due to hardware constraints we used we used a lower batch size compared to (Galkin et al., 2023), (Zhang et al., 2025) and (Huang et al., 2025) which might reduce the finetuned performance for Datasets trained with partial datasets.

## D   PRETRAINING SCALING

We investigated the scaling behavior of our approach with different sizes of the relational vocabulary. For each vocabulary set detailed above we conducted pretraining on each of the pretraining mixtures employed in (Galkin et al., 2023) (see Table 16). Compared to (Galkin et al., 2023) we had to decrease the batch sizes to be able to train all models with the same parameters.

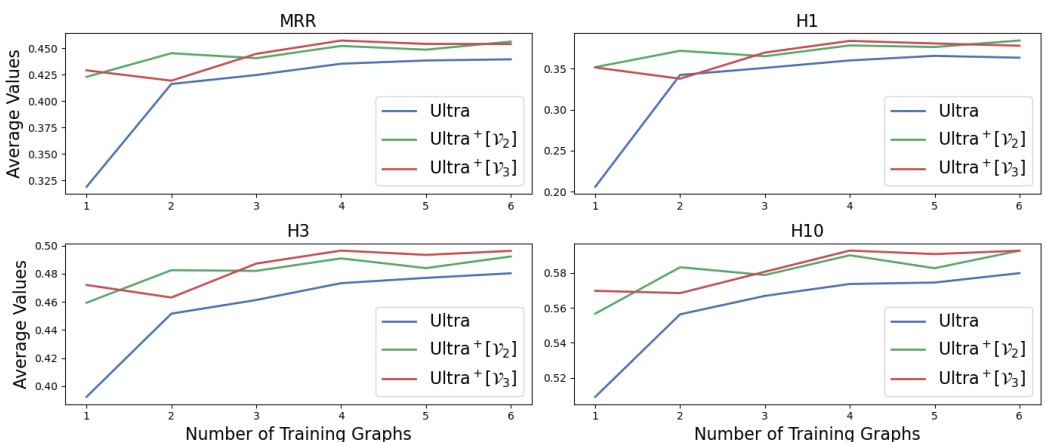

Figure 6: Average performance on 18 inductive (e) datasets of our $\mathrm{Ultra}^+$ models compared with Ultra, pretrained on 1 - 6 pretraining Graphs (see Table 16).

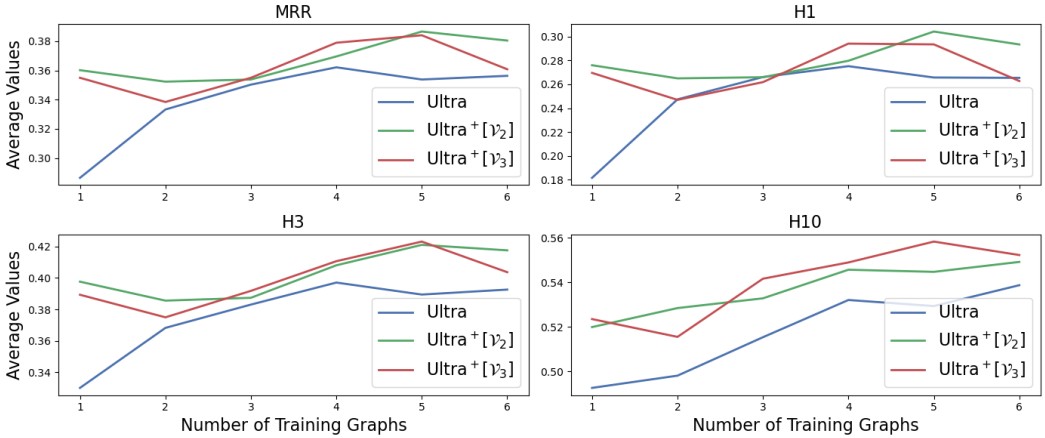

Figure 7: Average performance on 23 inductive (e,r) datasets of our $\mathrm{Ultra}^+$ models compared with Ultra, pretrained on 1 - 6 pretraining Graphs (see Table 16).

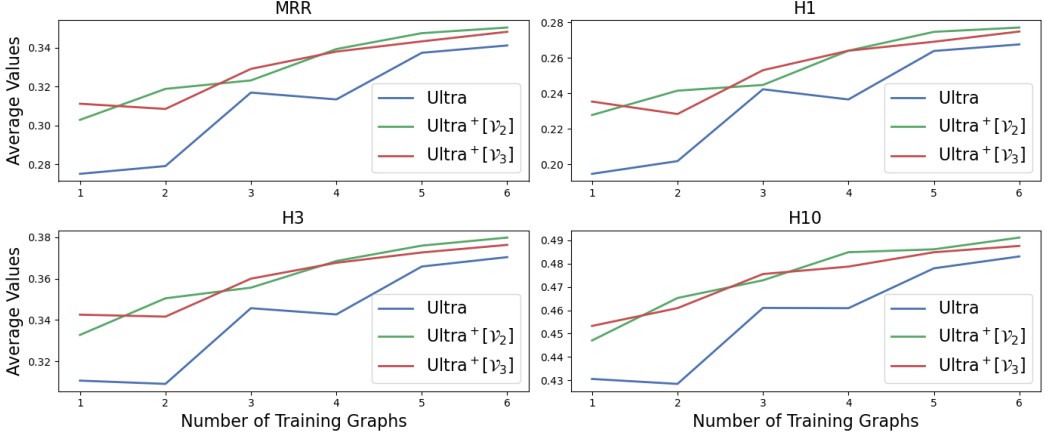

Figure 8: Average performance on 10 transductive datasets of our $\mathrm{Ultra}^+$ models compared with Ultra, pretrained on 1 - 6 pretraining Graphs (see Table 16).

Table 4: Statistics of **inductive** $(e, r)$ link prediction datasets. Triples are the number of edges given at training, validation, or test graphs, respectively, whereas Valid and Test denote triples to be predicted in the validation and test graphs.

| Dataset | Training Graph | | | Validation Graph | | | | Test Graph | | | |
|---|---|---|---|---|---|---|---|---|---|---|---|
| | Entities | Rels | Triples | Entities | Rels | Triples | Valid | Entities | Rels | Triples | Test |
| FB-25 | 5190 | 163 | 91571 | 4097 | 216 | 17147 | 5716 | 4097 | 216 | 17147 | 5716 |
| FB-50 | 5190 | 153 | 85375 | 4445 | 205 | 11636 | 3879 | 4445 | 205 | 11636 | 3879 |
| FB-75 | 4659 | 134 | 62809 | 2792 | 186 | 9316 | 3106 | 2792 | 186 | 9316 | 3106 |
| FB-100 | 4659 | 134 | 62809 | 2624 | 77 | 6987 | 2329 | 2624 | 77 | 6987 | 2329 |
| WK-25 | 12659 | 47 | 41873 | 3228 | 74 | 3391 | 1130 | 3228 | 74 | 3391 | 1131 |
| WK-50 | 12022 | 72 | 82481 | 9328 | 93 | 9672 | 3224 | 9328 | 93 | 9672 | 3225 |
| WK-75 | 6853 | 52 | 28741 | 2722 | 65 | 3430 | 1143 | 2722 | 65 | 3430 | 1144 |
| WK-100 | 9784 | 67 | 49875 | 12136 | 37 | 13487 | 4496 | 12136 | 37 | 13487 | 4496 |
| NL-0 | 1814 | 134 | 7796 | 2026 | 112 | 2287 | 763 | 2026 | 112 | 2287 | 763 |
| NL-25 | 4396 | 106 | 17578 | 2146 | 120 | 2230 | 743 | 2146 | 120 | 2230 | 744 |
| NL-50 | 4396 | 106 | 17578 | 2335 | 119 | 2576 | 859 | 2335 | 119 | 2576 | 859 |
| NL-75 | 2607 | 96 | 11058 | 1578 | 116 | 1818 | 606 | 1578 | 116 | 1818 | 607 |
| NL-100 | 1258 | 55 | 7832 | 1709 | 53 | 2378 | 793 | 1709 | 53 | 2378 | 793 |
| Metafam | 1316 | 28 | 13821 | 1316 | 28 | 13821 | 590 | 656 | 28 | 7257 | 184 |
| FBNELL | 4636 | 100 | 10275 | 4636 | 100 | 10275 | 1055 | 4752 | 183 | 10685 | 597 |
| Wiki MT1 tax | 10000 | 10 | 17178 | 10000 | 10 | 17178 | 1908 | 10000 | 9 | 16526 | 1834 |
| Wiki MT1 health | 10000 | 7 | 14371 | 10000 | 7 | 14371 | 1596 | 10000 | 7 | 14110 | 1566 |
| Wiki MT2 org | 10000 | 10 | 23233 | 10000 | 10 | 23233 | 2581 | 10000 | 11 | 21976 | 2441 |
| Wiki MT2 sci | 10000 | 16 | 16471 | 10000 | 16 | 16471 | 1830 | 10000 | 16 | 14852 | 1650 |
| Wiki MT3 art | 10000 | 45 | 27262 | 10000 | 45 | 27262 | 3026 | 10000 | 45 | 28023 | 3113 |
| Wiki MT3 infra | 10000 | 24 | 21990 | 10000 | 24 | 21990 | 2443 | 10000 | 27 | 21646 | 2405 |
| Wiki MT4 sci | 10000 | 42 | 12576 | 10000 | 42 | 12576 | 1397 | 10000 | 42 | 12516 | 1388 |
| Wiki MT4 health | 10000 | 21 | 15539 | 10000 | 21 | 15539 | 1725 | 10000 | 20 | 15337 | 1703 |

Table 5: Statistics of inductive $(e)$ link prediction datasets. Triples are the number of edges given at training, validation, or test graphs, respectively, whereas Valid and Test denote triples to be predicted in the validation and test graphs.

| Dataset | Rels | Training Graph | | Validation Graph | | | Test Graph | | |
|---|---|---|---|---|---|---|---|---|---|
| | | Entities | Triples | Entities | Triples | Valid | Entities | Triples | Test |
| FB-v1 | 180 | 1594 | 4245 | 1594 | 4245 | 489 | 1093 | 1993 | 411 |
| FB-v2 | 200 | 2608 | 9739 | 2608 | 9739 | 1166 | 1660 | 4145 | 947 |
| FB-v3 | 215 | 3668 | 17986 | 3668 | 17986 | 2194 | 2501 | 7406 | 1731 |
| FB-v4 | 219 | 4707 | 27203 | 4707 | 27203 | 3352 | 3051 | 11714 | 2840 |
| WN-v1 | 9 | 2746 | 5410 | 2746 | 5410 | 630 | 922 | 1618 | 373 |
| WN-v2 | 10 | 6954 | 15262 | 6954 | 15262 | 1838 | 2757 | 4011 | 852 |
| WN-v3 | 11 | 12078 | 25901 | 12078 | 25901 | 3097 | 5084 | 6327 | 1143 |
| WN-v4 | 9 | 3861 | 7940 | 3861 | 7940 | 934 | 7084 | 12334 | 2823 |
| NL-v1 | 14 | 3103 | 4687 | 3103 | 4687 | 414 | 225 | 833 | 201 |
| NL-v2 | 88 | 2564 | 8219 | 2564 | 8219 | 922 | 2086 | 4586 | 935 |
| NL-v3 | 142 | 4647 | 16393 | 4647 | 16393 | 1851 | 3566 | 8048 | 1620 |
| NL-v4 | 76 | 2092 | 7546 | 2092 | 7546 | 876 | 2795 | 7073 | 1447 |
| ILPC Small | 48 | 10230 | 78616 | 6653 | 20960 | 2908 | 6653 | 20960 | 2902 |
| ILPC Large | 65 | 46626 | 202446 | 29246 | 77044 | 10179 | 29246 | 77044 | 10184 |
| HM 1k | 11 | 36237 | 93364 | 36311 | 93364 | 1771 | 9899 | 18638 | 476 |
| HM 3k | 11 | 32118 | 71097 | 32250 | 71097 | 1201 | 19218 | 38285 | 1349 |
| HM 5k | 11 | 28601 | 57601 | 28744 | 57601 | 900 | 23792 | 48425 | 2124 |
| HM Indigo | 229 | 12721 | 121601 | 12797 | 121601 | 14121 | 14775 | 250195 | 14904 |

Table 6: Statistics of transductive link prediction datasets. Task denotes the prediction task: $h/t$ is predicting both heads and tails, and $t$ is predicting only tails.

| Dataset | Entities | Rels | Train | Valid | Test | Task |
|---|---|---|---|---|---|---|
| FB15k237 | 14541 | 237 | 272115 | 17535 | 20466 | $h/t$ |
| WN18RR | 40943 | 11 | 86835 | 3034 | 3134 | $h/t$ |
| CoDEx Small | 2034 | 42 | 32888 | 1827 | 1828 | $h/t$ |
| CoDEx Medium | 17050 | 51 | 185584 | 10310 | 10311 | $h/t$ |
| CoDEx Large | 77951 | 69 | 551193 | 30622 | 30622 | $h/t$ |
| NELL995 | 74536 | 200 | 149678 | 543 | 2818 | $h/t$ |
| YAGO310 | 123182 | 37 | 1079040 | 5000 | 5000 | $h/t$ |
| WDsinger | 10282 | 135 | 16142 | 2163 | 2203 | $h/t$ |
| NELL23k | 22925 | 200 | 25445 | 4961 | 4952 | $h/t$ |
| FB15k237(10) | 11512 | 237 | 27211 | 15624 | 18150 | $t$ |
| FB15k237(20) | 13166 | 237 | 54423 | 16963 | 19776 | $t$ |
| FB15k237(50) | 14149 | 237 | 136057 | 17449 | 20324 | $t$ |
| Hetionet | 45158 | 24 | 2025177 | 112510 | 112510 | $h/t$ |
| ConceptNet100k | 78334 | 34 | 100000 | 1200 | 1200 | $h/t$ |

Table 7: Full results for Ultra and Ultra$^+$ models on 10 transductive datasets. Baseline results are taken from (Huang et al., 2025).

| Model | Ultra | | Ultra$^+$ | | | | | | | | | | |
|---|---|---|---|---|---|---|---|---|---|---|---|---|---|
| Vocab. | $\mathcal{U}$ | | $\mathcal{V}_2^-$ | | $\mathcal{V}_2$ | | $\mathcal{V}_2^+$ | | $\mathcal{V}_3^-$ | | $\mathcal{V}_3$ | | $\mathcal{V}_3^+$ |
| Dataset | MRR | H10 | MRR | H10 | MRR | H10 | MRR | H10 | MRR | H10 | MRR | H10 | MRR | H10 |
| CoDExSmall | .479 | .668 | .469 | **.675** | **.486** | .675 | .480 | .675 | .475 | .667 | .484 | .674 | .475 | .671 |
| CoDExLarge | .339 | .466 | .343 | .470 | .342 | .471 | .336 | .466 | **.343** | **.473** | .340 | .468 | .335 | .462 |
| NELL995 | .444 | .583 | .461 | .598 | .446 | .600 | **.507** | **.644** | .472 | .601 | .451 | .613 | .484 | .637 |
| YAGO310 | .438 | .604 | .395 | .570 | .416 | .639 | .420 | .607 | .473 | .649 | **.505** | **.669** | .411 | .587 |
| WDsinger | .388 | .495 | .363 | .500 | .402 | .505 | .392 | **.512** | .388 | .503 | **.402** | .511 | .401 | .509 |
| NELL23k | .228 | .392 | .224 | .392 | .249 | .413 | .238 | .405 | .224 | .388 | **.250** | **.419** | .241 | .401 |
| FB15k237(10) | .237 | .403 | .245 | .400 | **.249** | **.404** | .244 | .395 | .233 | .382 | .245 | .400 | .240 | .390 |
| FB15k237(20) | .268 | .436 | .271 | .438 | **.274** | **.439** | .270 | .433 | .265 | .425 | .268 | .431 | .238 | .398 |
| FB15k237(50) | .323 | .525 | .325 | .525 | **.329** | **.527** | .324 | .526 | .323 | .519 | .326 | .524 | .313 | .502 |
| Hetionet | .287 | **.417** | .282 | .410 | **.301** | .417 | .259 | .381 | | | .278 | .405 | .280 | .390 |

# E DETAILS ON RELATION GRAPH COMPUTATION

## E.1 COMPLEXITY ANALYSIS

The time complexity of Ultra and Motif are computed in (Huang et al., 2025). This involves *(i)* estimating the computational complexity of generating the relation graph, by scanning the triples in the KGs and executing the sparse-matrix multiplication, *(ii)* in addition to applying a single forward pass in both the relation and entity encoders. These are

$$\mathcal{O}\left(\left(|\mathcal{E}|^2|\mathcal{R}| + |\mathcal{E}||\mathcal{R}|^2\right) + L\left(|\mathcal{R}|^2 d + |\mathcal{R}|d^2\right) + L\left(|\mathcal{T}|d + |\mathcal{E}|d^2\right)\right) \quad (2)$$

for Ultra and Motif equipped with 2-paths, and

$$\mathcal{O}\left(\left(|\mathcal{E}||\mathcal{R}|^3 + |\mathcal{E}|^2|\mathcal{R}|^2\right) + L\left(|\mathcal{R}|^3 d + |\mathcal{R}|d^2\right) + L\left(|\mathcal{T}|d + |\mathcal{E}|d^2\right)\right) \quad (3)$$

for Motif equipped with 3-paths. Ultra$^+$ replaces step *(i)* by running SPARQL ASK queries on the KGs. This implies we only need to scan the KGs without executing the SPMMs. Since the SPARQL queries in the vocabularies we defined are bounded in size and the filter conditions are simple equalities, their time complexity is $\mathcal{O}(1)$ (refer to Theorem 1 in (Pérez et al., 2006). Thus, we reduced the time complexity for 2-paths from $\mathcal{O}\left(|\mathcal{E}|^2|\mathcal{R}| + |\mathcal{E}||\mathcal{R}|^2\right)$ to $\mathcal{O}\left(|\mathcal{E}|^2|\mathcal{R}|\right)$ and for 3-paths from $\mathcal{O}\left(|\mathcal{E}||\mathcal{R}|^3 + |\mathcal{E}|^2|\mathcal{R}|^2\right)$ to $\mathcal{O}\left(|\mathcal{E}||\mathcal{R}|^3\right)$. As Ultra$^+$ constructs binary relation

Table 8: Full results for Ultra and Ultra$^+$ models on 23 inductive $(e, r)$ datasets. Baseline results are taken from (Huang et al., 2025).

| Model | Ultra | | Ultra$^+$ | | | | | | | | | | | |
| --- | --- | --- | --- | --- | --- | --- | --- | --- | --- | --- | --- | --- | --- | --- |
| Vocab. | $\mathcal{U}$ | | $\mathcal{V}_2^-$ | | $\mathcal{V}_2$ | | $\mathcal{V}_2^+$ | | $\mathcal{V}_3^-$ | | $\mathcal{V}_3$ | | $\mathcal{V}_3^+$ | |
| Dataset | MRR | H10 | MRR | H10 | MRR | H10 | MRR | H10 | MRR | H10 | MRR | H10 | MRR | H10 |
| FB-25 | .385 | .636 | .386 | .639 | **.396** | .639 | .394 | **.647** | .384 | .638 | .393 | .643 | .391 | .645 |
| FB-50 | .332 | .535 | .329 | .540 | .339 | .548 | .339 | **.551** | .330 | .543 | .341 | .546 | **.343** | .547 |
| FB-75 | .397 | .596 | **.404** | **.609** | .404 | .605 | .403 | .607 | .399 | .604 | .404 | .605 | .398 | .603 |
| FB-100 | .443 | .626 | .435 | .627 | **.443** | .625 | .439 | .633 | .438 | .628 | .443 | **.641** | .431 | .638 |
| WK-25 | .301 | .505 | .284 | .488 | **.324** | **.530** | .280 | .486 | .293 | .505 | .304 | .491 | .305 | .503 |
| WK-50 | .157 | .305 | .130 | .287 | **.174** | **.321** | .168 | .315 | .159 | .285 | .168 | .319 | .166 | .307 |
| WK-75 | .375 | **.538** | .373 | .517 | **.380** | .537 | .367 | .516 | .364 | .533 | .371 | .533 | .374 | .524 |
| WK-100 | .180 | .298 | .169 | .294 | .180 | .302 | .175 | .294 | .176 | .291 | .173 | .291 | **.185** | **.307** |
| NL-0 | .334 | .510 | .318 | .502 | .367 | .551 | .336 | .525 | .303 | .505 | **.370** | **.566** | .364 | .546 |
| NL-25 | .373 | .544 | .313 | .495 | .370 | .552 | .349 | .507 | .358 | .542 | .349 | **.585** | **.391** | .573 |
| NL-50 | .389 | .536 | .358 | .531 | **.406** | **.579** | .349 | .538 | .364 | .554 | .382 | .563 | .393 | .568 |
| NL-75 | .336 | .528 | .307 | .487 | .348 | .529 | .302 | .495 | .316 | .490 | .348 | .539 | **.349** | **.546** |
| NL-100 | .442 | .636 | .401 | .627 | .477 | .681 | .449 | .660 | .444 | .657 | .479 | **.694** | **.483** | .690 |
| Metafam | .428 | .739 | .156 | .503 | .262 | .723 | **.484** | **.962** | .173 | .560 | .279 | .851 | .310 | .872 |
| FBNELL | .461 | .631 | .463 | .634 | .484 | .659 | .482 | .652 | .471 | .640 | .492 | .647 | **.496** | **.679** |
| Wiki MT1 tax | .240 | .306 | .150 | .300 | .260 | **.436** | .251 | .311 | .234 | .347 | **.286** | .433 | .238 | .305 |
| Wiki MT1 health | .327 | .430 | .291 | .394 | .362 | .432 | .312 | .400 | .373 | .457 | **.375** | **.458** | .363 | .449 |
| Wiki MT2 org | .089 | .152 | .096 | .157 | **.098** | .158 | .091 | .156 | .096 | .159 | .098 | **.163** | .096 | .159 |
| Wiki MT2 sci | .263 | .415 | .262 | .387 | .283 | .450 | .283 | .424 | .266 | .427 | **.300** | **.458** | .270 | .380 |
| Wiki MT3 art | .262 | .413 | .272 | .420 | .276 | .422 | .277 | .429 | .277 | .429 | **.286** | **.435** | .278 | .420 |
| Wiki MT3 infra | .634 | .769 | .647 | **.791** | .637 | .783 | .640 | .774 | .624 | .755 | **.647** | .782 | .638 | .765 |
| Wiki MT4 sci | .285 | .449 | .295 | .469 | .301 | .464 | .294 | .466 | **.301** | .465 | .295 | **.471** | .296 | .463 |
| Wiki MT4 health | **.625** | **.755** | .595 | .746 | .568 | .729 | .558 | .723 | .598 | .729 | .583 | .744 | .619 | .746 |

Table 9: Full results for Ultra and Ultra$^+$ models on 18 inductive $(e)$ datasets. Baseline results are taken from (Huang et al., 2025).

| Model | Ultra | | Ultra$^+$ | | | | | | | | | | | |
| --- | --- | --- | --- | --- | --- | --- | --- | --- | --- | --- | --- | --- | --- | --- |
| Vocab. | $\mathcal{U}$ | | $\mathcal{V}_2^-$ | | $\mathcal{V}_2$ | | $\mathcal{V}_2^+$ | | $\mathcal{V}_3^-$ | | $\mathcal{V}_3$ | | $\mathcal{V}_3^+$ | |
| Dataset | MRR | H10 | MRR | H10 | MRR | H10 | MRR | H10 | MRR | H10 | MRR | H10 | MRR | H10 |
| FB-v1 | .498 | .653 | .468 | .656 | .498 | .661 | .477 | .670 | .492 | **.687** | **.503** | .663 | .503 | .678 |
| FB-v2 | .504 | .695 | .502 | .707 | .510 | .703 | .512 | .697 | .507 | **.718** | **.529** | .716 | .525 | .712 |
| FB-v3 | .489 | .656 | .479 | .648 | .488 | .650 | .489 | .661 | .488 | .654 | **.497** | .660 | .494 | **.661** |
| FB-v4 | .478 | .665 | .477 | .675 | .488 | .675 | .485 | .678 | .474 | .670 | **.489** | **.679** | .489 | .677 |
| WN-v1 | .658 | .764 | .203 | .555 | **.705** | .792 | .697 | .796 | .655 | .763 | .703 | **.811** | .690 | .794 |
| WN-v2 | .648 | .749 | .642 | .762 | **.698** | .786 | .455 | .741 | .637 | .743 | .676 | .783 | .687 | **.789** |
| WN-v3 | .367 | .464 | .387 | .505 | .361 | .514 | .358 | .523 | .373 | .479 | **.416** | .539 | .413 | **.542** |
| WN-v4 | .603 | .704 | .598 | .711 | .651 | .730 | .568 | .718 | .605 | .711 | **.657** | **.738** | .644 | .723 |
| NL-v1 | .694 | .896 | .524 | .771 | .739 | .920 | .583 | .866 | .597 | .644 | **.749** | **.930** | .585 | .866 |
| NL-v2 | .516 | .715 | .507 | .699 | .551 | .728 | .550 | .750 | .528 | .735 | .565 | .754 | **.572** | **.763** |
| NL-v3 | .510 | .690 | .493 | .666 | .550 | .728 | .548 | .729 | .526 | .723 | **.562** | .737 | .560 | **.750** |
| NL-v4 | .483 | .704 | .491 | .715 | .505 | .728 | .517 | .746 | .505 | .730 | .510 | .737 | **.521** | **.756** |
| ILPC small | .296 | .445 | **.304** | .450 | .299 | .450 | .295 | .454 | .303 | .447 | .301 | .450 | .304 | **.456** |
| ILPC large | .292 | .417 | **.305** | **.427** | .287 | .423 | .290 | .426 | .299 | .424 | .297 | .419 | .297 | .423 |
| HM 1k | .058 | .122 | .064 | .126 | .065 | .122 | **.079** | **.147** | .079 | .132 | .042 | .068 | .036 | .076 |
| HM 3k | .055 | .112 | .048 | .095 | .046 | .079 | **.065** | **.116** | .056 | .101 | .039 | .063 | .034 | .078 |
| HM 5k | .051 | .103 | .045 | .091 | .043 | .080 | **.057** | **.104** | .051 | .093 | .032 | .054 | .030 | .070 |
| HM indigo | .446 | .649 | .439 | .649 | .446 | .649 | **.449** | **.654** | .432 | .644 | .440 | .651 | .438 | .650 |

graphs, Ultra and Ultra$^+$ have the same forward pass time complexity for both 2- and 3-paths. In overall,

$$\mathcal{O}\left(|\mathcal{E}|^2|\mathcal{R}| + L\left(|\mathcal{R}|^2 d + |\mathcal{R}| d^2\right) + L\left(|\mathcal{T}| d + |\mathcal{E}| d^2\right)\right) \qquad (4)$$

Table 10: Finetuned Inductive $(e, r)$ Performance Comparison

| Dataset | Ultra | | Motif | | Ultra$^+[\mathcal{V}_3^+]$ | |
|---|---|---|---|---|---|---|
| | MRR | H@10 | MRR | H@10 | MRR | H@10 |
| FB-25 | 0.383 | 0.635 | 0.388 | 0.635 | **0.391** | **0.642** |
| FB-50 | 0.334 | 0.538 | **0.340** | **0.544** | 0.333 | 0.541 |
| FB-75 | 0.400 | 0.598 | 0.399 | **0.607** | **0.403** | 0.604 |
| FB-100 | 0.444 | **0.643** | 0.439 | 0.642 | **0.445** | 0.640 |
| WK-25 | **0.321** | **0.535** | 0.317 | 0.505 | 0.298 | 0.487 |
| WK-50 | 0.140 | 0.280 | 0.160 | 0.304 | **0.162** | **0.314** |
| WK-75 | 0.380 | 0.530 | 0.371 | **0.535** | **0.387** | 0.529 |
| WK-100 | 0.168 | 0.286 | 0.173 | 0.284 | **0.180** | **0.294** |
| NL-0 | **0.329** | 0.551 | 0.328 | **0.556** | 0.305 | 0.490 |
| NL-25 | **0.407** | **0.596** | 0.390 | 0.580 | 0.353 | 0.540 |
| NL-50 | **0.418** | **0.595** | 0.414 | 0.573 | 0.399 | 0.579 |
| NL-75 | **0.374** | **0.570** | 0.360 | 0.548 | 0.360 | 0.563 |
| NL-100 | 0.458 | **0.684** | 0.464 | 0.682 | **0.477** | 0.661 |
| Metafam | 0.997 | **1.000** | **1.000** | **1.000** | **1.000** | **1.000** |
| FBNELL | **0.481** | 0.661 | **0.481** | **0.664** | 0.445 | 0.626 |
| MT1-tax | 0.330 | 0.459 | 0.416 | 0.522 | **0.429** | **0.533** |
| MT1-health | 0.380 | 0.467 | 0.385 | **0.473** | **0.386** | 0.462 |
| MT2-org | 0.104 | 0.170 | **0.106** | 0.170 | 0.104 | **0.175** |
| MT2-sci | 0.311 | 0.451 | **0.326** | **0.520** | 0.320 | 0.427 |
| MT3-art | 0.306 | 0.473 | **0.315** | 0.469 | **0.315** | **0.479** |
| MT3-infra | 0.657 | 0.807 | **0.683** | **0.827** | **0.683** | 0.821 |
| MT4-sci | 0.303 | 0.478 | 0.309 | 0.483 | **0.311** | **0.489** |
| MT4-health | 0.704 | 0.785 | 0.703 | 0.787 | **0.709** | **0.788** |

Table 11: Finetuned Inductive $(e)$ Performance Comparison

| Dataset | Ultra | | Motif | | Ultra$^+[\mathcal{V}_3^+]$ | |
|---|---|---|---|---|---|---|
| | MRR | H@10 | MRR | H@10 | MRR | H@10 |
| FB-v1 | 0.509 | 0.670 | **0.530** | **0.702** | 0.510 | 0.669 |
| FB-v2 | 0.524 | 0.710 | **0.557** | **0.744** | 0.540 | 0.730 |
| FB-v3 | 0.504 | 0.663 | **0.519** | **0.684** | 0.509 | 0.665 |
| FB-v4 | 0.496 | 0.684 | **0.508** | **0.695** | 0.497 | 0.683 |
| WN-v1 | 0.685 | 0.793 | 0.703 | **0.806** | **0.705** | 0.789 |
| WN-v2 | 0.679 | 0.779 | 0.680 | **0.781** | **0.694** | 0.779 |
| WN-v3 | 0.411 | 0.546 | **0.466** | **0.590** | 0.433 | 0.550 |
| WN-v4 | 0.614 | 0.720 | 0.659 | 0.733 | **0.662** | **0.748** |
| NL-v1 | 0.757 | 0.878 | 0.712 | 0.873 | **0.811** | **0.931** |
| NL-v2 | **0.575** | 0.761 | 0.566 | **0.765** | 0.572 | 0.761 |
| NL-v3 | 0.563 | 0.755 | 0.580 | 0.764 | **0.589** | **0.781** |
| NL-v4 | 0.469 | 0.733 | 0.507 | 0.740 | **0.537** | **0.761** |
| ILPC Small | 0.303 | **0.453** | 0.302 | 0.449 | **0.307** | 0.450 |
| ILPC Large | **0.308** | 0.431 | 0.307 | **0.432** | 0.313 | **0.435** |
| HM-1k | 0.042 | 0.100 | **0.067** | **0.107** | 0.043 | 0.098 |
| HM-3k | 0.030 | 0.090 | **0.054** | **0.103** | 0.025 | 0.062 |
| HM-5k | 0.025 | 0.068 | **0.049** | **0.091** | 0.025 | 0.050 |
| HM-Indigo | **0.432** | **0.639** | 0.426 | 0.635 | 0.393 | 0.520 |

Table 12: Finetuned Transductive Performance Comparison

| Dataset | Ultra | | Motif | | Ultra$^+[\mathcal{V}_3^+]$ | |
|---|---|---|---|---|---|---|
| | MRR | H@10 | MRR | H@10 | MRR | H@10 |
| CoDEx Small | 0.490 | **0.686** | 0.490 | 0.680 | **0.496** | 0.684 |
| CoDEx Large | 0.343 | 0.478 | 0.355 | 0.489 | **0.359** | **0.495** |
| NELL-995 | 0.509 | 0.660 | 0.514 | 0.655 | **0.547** | **0.678** |
| YAGO310 | 0.557 | 0.710 | 0.603 | **0.735** | **0.607** | **0.735** |
| WDsinger | 0.417 | 0.526 | 0.423 | 0.532 | **0.431** | **0.535** |
| NELL23k | 0.268 | 0.450 | 0.256 | 0.441 | **0.270** | **0.453** |
| FB15k237(10) | 0.254 | 0.411 | 0.254 | 0.411 | **0.263** | **0.416** |
| FB15k237(20) | 0.274 | **0.445** | 0.273 | 0.444 | **0.281** | **0.445** |
| FB15k237(50) | 0.325 | 0.528 | 0.323 | 0.523 | **0.335** | **0.531** |
| Hetionet | 0.399 | 0.538 | 0.446 | 0.575 | **0.464** | **0.599** |

Table 13: Average finetuned link prediction MRR and H10 over 51 KGs. Baseline results are taken from (Huang et al., 2025). $\mathcal{P}_n$, O and C stand for n-, open, and closed paths; and N-M stands for many-to-many subgraphs

| Model | Structural Vocabulary | | | Ind.$(e)$ (18 KGs) | | Ind.$(e,r)$ (23 KGs) | | Transd. (10 KGs) | | Total Avg. (51 KGs) | |
|---|---|---|---|---|---|---|---|---|---|---|---|
| | $\mathcal{V}$ | Definition | #$\mathcal{V}$ | MRR | H10 | MRR | H10 | MRR | H10 | MRR | H10 |
| Ultra | $\mathcal{U}_2$ | $\mathcal{P}_2$ | 4 | .442 | .582 | .397 | .556 | .384 | .543 | .410 | .563 |
| Motif | $\mathcal{U}_3$ | $\mathcal{P}_3$ | 12 | **.455** | **.594** | **.401** | **.558** | .394 | .549 | .419 | **.569** |
| Ultra$^+$ | $\mathcal{V}_3$ | O. & C. $\mathcal{P}_3$ | 24 | **.455** | .581 | .400 | .551 | **.405** | **.557** | **.420** | .563 |

for Ultra$^+$ equipped with 2-paths, and

$$\mathcal{O}\left(\mid \mathcal{E} \mid\mid \mathcal{R} \mid^3 + L\left(\mid \mathcal{R} \mid^2 d + \mid \mathcal{R} \mid d^2\right) + L\left(\mid \mathcal{T} \mid d + \mid \mathcal{E} \mid d^2\right)\right) \qquad (5)$$

for Ultra$^+$ equipped with 3-paths.

### E.2 EXPERIMENTAL ANALYSIS

In Table 17, we compare the computation of relation graphs based on sparse matrix multiplication with query-based computation.

**Implementation and Experiment Details** Batching is used in the implementation of the SPMM-based computation of the relation graph, since the intermediate matrix products are too large to fit into memory. Further improvements to our implementation are possible, but not to the extent that the computation will reach the speed of Query based computations.

The Query based relation graph computation was implemented using rdflib (Krech et al., 2025) and the oxrdflib extension based on oxigraph (Pellissier Tanon) which provides efficient SPARQL query resolution and supports large datasets.

This comparison has been executed on a Machine with 64cpu (2 * 32 core AMD EPYC) cores 256GB RAM and an Nvidia A100 (80GB) GPU.

### E.3 SPMM FORMULATION OF Ultra$^+$ VOCABULARY

The adjacency matrices for the relation graphs in (Galkin et al., 2023) and (Huang et al., 2025) had convenient representaitons in terms of Sparse Matrix Multiplications (SPMM) expressed as products of the adjacency matrix $A \in \mathbb{R}^{n \times n}$ of the KG and two matrices $\mathbb{E}_h \in \mathbb{R}^{n \times m}$ and $E_t \in \mathbb{R}^{m \times n}$. The following formulas are used to construct the binary edges of the relation graph in Motif (see Section F in (Huang et al., 2025)) and Ultra (see Section B in (Galkin et al., 2023)):

Table 14: Hyperparameters for fine-tuning $\mathrm{Ultra}^{+}$. Full represents a whole epoch with the entire dataset being used

| Datasets | Epoch | Batch per Epoch |
|---|---|---|
| FB 25-100 | 3 | full |
| WK 25-100 | 3 | full |
| NL 0-100 | 3 | full |
| MT1-MT4 | 3 | full |
| Metafam, FBNELL | 3 | full |
| FB v1-v4 | 1 | full |
| WN v1-v4 | 1 | full |
| NL v1-v4 | 3 | full |
| ILPC Small | 3 | full |
| ILPC Large | 1 | 1000 |
| HM 1k-5k, Indigo | 1 | 100 |
| FB15k237 | 1 | full |
| WN18RR | 1 | full |
| CoDEx Small | 1 | 4000 |
| CoDEx Medium | 1 | 4000 |
| CoDEx Large | 1 | 2000 |
| NELL-995 | 1 | full |
| YAGO310 | 1 | 2000 |
| WDsinger | 3 | full |
| NELL23k | 3 | full |
| FB15k237(10) | 1 | full |
| FB15k237(20) | 1 | full |
| FB15k237(50) | 1 | 1000 |
| Hetionet | 1 | 4000 |

Table 15: Global hyper-parameters for fine-tuning.

| Hyperparameter | Value |
|---|---|
| Optimizer | AdamW |
| Learning rate | 0.0005 |
| Adversarial temperature | 1 |
| # Negatives | 256 |
| Batch size | 8 |
| # Repetitions | 1 |

Table 16: Graphs and training parameters in different pre-training mixtures in Figures 6, 7 and 8

| | 1 | 2 | 3 | 4 | 5 | 6 |
|---|---|---|---|---|---|---|
| FB15k237 | ✓ | ✓ | ✓ | ✓ | ✓ | ✓ |
| WN18RR | | ✓ | ✓ | ✓ | ✓ | ✓ |
| CoDEx-M | | | ✓ | ✓ | ✓ | ✓ |
| NELL995 | | | | ✓ | ✓ | ✓ |
| YAGO 310 | | | | | ✓ | ✓ |
| ConceptNet100k | | | | | | ✓ |
| Batch size | 32 | 16 | 16 | 16 | 8 | 8 |
| # steps | 200,000 | 400,000 | 300,000 | 400,000 | 200,000 | 200,000 |

$$\boldsymbol{A}_{h2h} = \mathrm{spmm}\left(\boldsymbol{E}_h^T, \boldsymbol{E}_h\right) \in \mathbb{R}^{m \times m}$$

$$\boldsymbol{A}_{t2t} = \mathrm{spmm}\left(\boldsymbol{E}_t^T, \boldsymbol{E}_t\right) \in \mathbb{R}^{m \times m}$$

$$\boldsymbol{A}_{h2t} = \mathrm{spmm}\left(\boldsymbol{E}_h^T, \boldsymbol{E}_t\right) \in \mathbb{R}^{m \times m}$$

$$\boldsymbol{A}_{t2h} = \mathrm{spmm}\left(\boldsymbol{E}_t^T, \boldsymbol{E}_h\right) \in \mathbb{R}^{m \times m}$$

Table 17: Runtime and memory usage comparison for relation graph computation using $\mathcal{V}_2$ with the formulation in E.3 and the Query based computation. Time is displayed in hours:minutes:seconds

| Dataset | Query based | | | SPMM | | |
|---|---|---|---|---|---|---|
| | time | RAM usage | VRAM usage | time | RAM | VRAM (GPU) |
| WM18RR | 00:00:08 | 10 GB | – | 00:10:24 | 10 GB | 50 GB |
| FB15k237 | 00:01:03 | 23 GB | – | 01:52:43 | 40 GB | 50 GB |
| CODEX Medium | 00:00:30 | 12 GB | – | 01:07:52 | 10 GB | 50 GB |

and the 3-ary edges in Motif:

$$\boldsymbol{A}_{hfh} = \mathrm{spmm}\left(\boldsymbol{E}_h^T, \boldsymbol{A}, \boldsymbol{E}_h\right) \in \mathbb{R}^{m \times m \times m}$$
$$\boldsymbol{A}_{tft} = \mathrm{spmm}\left(\boldsymbol{E}_t^T, \boldsymbol{A}, \boldsymbol{E}_t\right) \in \mathbb{R}^{m \times m \times m}$$
$$\boldsymbol{A}_{hft} = \mathrm{spmm}\left(\boldsymbol{E}_h^T, \boldsymbol{A}, \boldsymbol{E}_t\right) \in \mathbb{R}^{m \times m \times m}$$
$$\boldsymbol{A}_{tfh} = \mathrm{spmm}\left(\boldsymbol{E}_t^T, \boldsymbol{A}, \boldsymbol{E}_h\right) \in \mathbb{R}^{m \times m \times m}.$$

The resulting adjacency matrices fail to distinguish between loops and paths that go through the same entity multiple times. One way to enable these models to distinguish between closed and open two-paths is to use the full product with masking. For a graphlet $g = uv_z$ in the vocabulary $\mathcal{V}_2$ where $u, v \in \{f, r\}$ and $z \in \{0, c\}$ the adjacency matrix is given by $A_g = (a_{g,ij})_{1 \leq i,j \leq m} \in \mathbb{R}^{m \times m \times |\mathcal{V}_2|}$

$$a_{uv_o,ij} = \sum_{\substack{l,m,n \\ l \neq m, m \neq n, n \neq l}} \tau(u, A^i)_{lm} \cdot \tau(v, A^j)_{mn}, \tag{6}$$

$$a_{uv_c,ij} = \sum_{\substack{l,m \\ l \neq m}} \tau(u, A^i)_{lm} \cdot \tau(v, A^j)_{ml} \ \text{ with } u, v \in \{f, r\} \tag{7}$$

where $\tau : \{r, f\} \times \mathbb{R}^{n \times n} \to \mathbb{R}^{n \times n}$:

$$\tau(u, A) := \begin{cases} A; & u = f \\ A^T; & u = r. \end{cases}$$

Similar equations hold for longer path.

### E.4 SPARQL QUERIES

We display the query patterns for the Vocabularies employed in our Experiments in Tables 18-20

Table 18: 3-Relation Pattern SPARQL Queries correspinding to vocabulary $\mathcal{V}_3$

| Pattern | SPARQL Query |
|---|---|
| fffo | ```ASK WHERE {
  ?e0 {rel1} ?e1 .
  ?e1 ?rel_0 ?e2 .
  ?e2 {rel2} ?e3 .
  FILTER(?e0 != ?e1 && ?e0 != ?e2 &&
         ?e1 != ?e2 && ?e1 != ?e3 &&
         ?e2 != ?e3 && ?e0 != ?e3)
}``` |
| fffc | ```ASK WHERE {
  ?e0 {rel1} ?e1 .
  ?e1 ?rel_0 ?e2 .
  ?e2 {rel2} ?e0 .
  FILTER(?e0 != ?e1 && ?e1 != ?e2 &&
         ?e0 != ?e2)
}``` |
| ffro | ```ASK WHERE {
  ?e0 {rel1} ?e1 .
  ?e1 ?rel_0 ?e2 .
  ?e3 {rel2} ?e2 .
  FILTER(?e0 != ?e1 && ?e0 != ?e2 &&
         ?e1 != ?e2 && ?e1 != ?e3 &&
         ?e2 != ?e3 && ?e0 != ?e3)
}``` |
| ffrc | ```ASK WHERE {
  ?e0 {rel1} ?e1 .
  ?e1 ?rel_0 ?e2 .
  ?e0 {rel2} ?e2 .
  FILTER(?e0 != ?e1 && ?e1 != ?e2 &&
         ?e0 != ?e2)
}``` |
| frfo | ```ASK WHERE {
  ?e0 {rel1} ?e1 .
  ?e2 ?rel_0 ?e1 .
  ?e2 {rel2} ?e3 .
  FILTER(?e0 != ?e1 && ?e0 != ?e2 &&
         ?e1 != ?e2 && ?e1 != ?e3 &&
         ?e2 != ?e3 && ?e0 != ?e3)
}``` |
| | Continued on next page |

**Table 18 – continued from previous page**

| Pattern | SPARQL Query |
|---|---|
| frfc | <pre>ASK WHERE {
  ?e0 {rel1} ?e1 .
  ?e2 ?rel_0 ?e1 .
  ?e2 {rel2} ?e0 .
  FILTER(?e0 != ?e1 && ?e1 != ?e2 &&
         ?e0 != ?e2)
}</pre> |
| frro | <pre>ASK WHERE {
  ?e0 {rel1} ?e1 .
  ?e2 ?rel_0 ?e1 .
  ?e3 {rel2} ?e2 .
  FILTER(?e0 != ?e1 && ?e0 != ?e2 &&
         ?e1 != ?e2 && ?e1 != ?e3 &&
         ?e2 != ?e3 && ?e0 != ?e3)
}</pre> |
| frrc | <pre>ASK WHERE {
  ?e0 {rel1} ?e1 .
  ?e2 ?rel_0 ?e1 .
  ?e0 {rel2} ?e2 .
  FILTER(?e0 != ?e1 && ?e1 != ?e2 &&
         ?e0 != ?e2)
}</pre> |
| rffo | <pre>ASK WHERE {
  ?e1 {rel1} ?e0 .
  ?e1 ?rel_0 ?e2 .
  ?e2 {rel2} ?e3 .
  FILTER(?e0 != ?e1 && ?e0 != ?e2 &&
         ?e1 != ?e2 && ?e1 != ?e3 &&
         ?e2 != ?e3 && ?e0 != ?e3)
}</pre> |
| rffc | <pre>ASK WHERE {
  ?e1 {rel1} ?e0 .
  ?e1 ?rel_0 ?e2 .
  ?e2 {rel2} ?e0 .
  FILTER(?e0 != ?e1 && ?e1 != ?e2 &&
         ?e0 != ?e2)
}</pre> |
| | Continued on next page |

**Table 18 – continued from previous page**

| Pattern | SPARQL Query |
|---------|--------------|
| rfro | ```
ASK WHERE {
  ?e1 {rel1} ?e0 .
  ?e1 ?rel_0 ?e2 .
  ?e3 {rel2} ?e2 .
  FILTER(?e0 != ?e1 && ?e0 != ?e2 &&
         ?e1 != ?e2 && ?e1 != ?e3 &&
         ?e2 != ?e3 && ?e0 != ?e3)
}
``` |
| rfrc | ```
ASK WHERE {
  ?e1 {rel1} ?e0 .
  ?e1 ?rel_0 ?e2 .
  ?e0 {rel2} ?e2 .
  FILTER(?e0 != ?e1 && ?e1 != ?e2 &&
         ?e0 != ?e2)
}
``` |
| rrfo | ```
ASK WHERE {
  ?e1 {rel1} ?e0 .
  ?e2 ?rel_0 ?e1 .
  ?e2 {rel2} ?e3 .
  FILTER(?e0 != ?e1 && ?e0 != ?e2 &&
         ?e1 != ?e2 && ?e1 != ?e3 &&
         ?e2 != ?e3 && ?e0 != ?e3)
}
``` |
| rrfc | ```
ASK WHERE {
  ?e1 {rel1} ?e0 .
  ?e2 ?rel_0 ?e1 .
  ?e2 {rel2} ?e0 .
  FILTER(?e0 != ?e1 && ?e1 != ?e2 &&
         ?e0 != ?e2)
}
``` |
| rrro | ```
ASK WHERE {
  ?e1 {rel1} ?e0 .
  ?e2 ?rel_0 ?e1 .
  ?e3 {rel2} ?e2 .
  FILTER(?e0 != ?e1 && ?e0 != ?e2 &&
         ?e1 != ?e2 && ?e1 != ?e3 &&
         ?e2 != ?e3 && ?e0 != ?e3)
}
``` |
| | Continued on next page |

**Table 18 – continued from previous page**

| Pattern | SPARQL Query |
|---------|--------------|
| rrrc | <pre>ASK WHERE {
  ?e1 {rel1} ?e0 .
  ?e2 ?rel_0 ?e1 .
  ?e0 {rel2} ?e2 .
  FILTER(?e0 != ?e1 && ?e1 != ?e2 &&
         ?e0 != ?e2)
}</pre> |

Table 19: 2-Relation Pattern SPARQL Queries corresponding to vocabulary $\mathcal{V}_2$

| Pattern | SPARQL Query |
|---------|--------------|
| ffo | <pre>ASK WHERE {
  ?e0 {rel1} ?e1 .
  ?e1 {rel2} ?e2 .
  FILTER(?e0 != ?e1 && ?e1 != ?e2 &&
         ?e0 != ?e2)
}</pre> |
| ffc | <pre>ASK WHERE {
  ?e0 {rel1} ?e1 .
  ?e1 {rel2} ?e0 .
  FILTER(?e0 != ?e1)
}</pre> |
| fro | <pre>ASK WHERE {
  ?e0 {rel1} ?e1 .
  ?e2 {rel2} ?e1 .
  FILTER(?e0 != ?e1 && ?e1 != ?e2 &&
         ?e0 != ?e2)
}</pre> |
| frc | <pre>ASK WHERE {
  ?e0 {rel1} ?e1 .
  ?e0 {rel2} ?e1 .
  FILTER(?e0 != ?e1)
}</pre> |
| rfo | <pre>ASK WHERE {
  ?e1 {rel1} ?e0 .
  ?e1 {rel2} ?e2 .
  FILTER(?e0 != ?e1 && ?e1 != ?e2 &&
         ?e0 != ?e2)
}</pre> |

**Table 19 – continued from previous page**

| Pattern | SPARQL Query |
|---------|--------------|
| rfc | ```
ASK WHERE {
  ?e1 {rel1} ?e0 .
  ?e1 {rel2} ?e0 .
  FILTER(?e0 != ?e1)
}
``` |
| rro | ```
ASK WHERE {
  ?e1 {rel1} ?e0 .
  ?e2 {rel2} ?e1 .
  FILTER(?e0 != ?e1 && ?e1 != ?e2 &&
         ?e0 != ?e2)
}
``` |
| rrc | ```
ASK WHERE {
  ?e1 {rel1} ?e0 .
  ?e0 {rel2} ?e1 .
  FILTER(?e0 != ?e1)
}
``` |

Table 20: N-M Pattern SPARQL Queries corresponding to vocabulary $\mathcal{V}_\bullet^+$

| Pattern | SPARQL Query |
|---------|--------------|
| ffo_1-2 | ```
ASK WHERE {
  ?e0 {rel1} ?e1 .
  ?e1 {rel2} ?e2 .
  ?e1 {rel2} ?e3 .
  FILTER(?e0 != ?e1 && ?e1 != ?e2 &&
         ?e2 != ?e3 && ?e3 != ?e0 &&
         ?e0 != ?e2 && ?e1 != ?e2)
}
``` |
| fro_1-2 | ```
ASK WHERE {
  ?e0 {rel1} ?e1 .
  ?e2 {rel2} ?e1 .
  ?e3 {rel2} ?e1 .
  FILTER(?e0 != ?e1 && ?e1 != ?e2 &&
         ?e2 != ?e3 && ?e3 != ?e0 &&
         ?e0 != ?e2 && ?e1 != ?e2)
}
``` |
| | Continued on next page |

**Table 20 – continued from previous page**

| Pattern | SPARQL Query |
|---------|--------------|
| rfo_1-2 | ```
ASK WHERE {
  ?e1 {rel1} ?e0 .
  ?e1 {rel2} ?e2 .
  ?e1 {rel2} ?e3 .
  FILTER(?e0 != ?e1 && ?e1 != ?e2 &&
         ?e2 != ?e3 && ?e3 != ?e0 &&
         ?e0 != ?e2 && ?e1 != ?e2)
}
``` |
| rro_1-2 | ```
ASK WHERE {
  ?e1 {rel1} ?e0 .
  ?e2 {rel2} ?e1 .
  ?e3 {rel2} ?e1 .
  FILTER(?e0 != ?e1 && ?e1 != ?e2 &&
         ?e2 != ?e3 && ?e3 != ?e0 &&
         ?e0 != ?e2 && ?e1 != ?e2)
}
``` |
| ffo_2-2 | ```
ASK WHERE {
  ?e0 {rel1} ?e2 .
  ?e1 {rel1} ?e2 .
  ?e2 {rel2} ?e3 .
  ?e2 {rel2} ?e4 .
  FILTER(?e0 != ?e1 && ?e1 != ?e2 &&
         ?e2 != ?e3 && ?e3 != ?e0 &&
         ?e0 != ?e2 && ?e1 != ?e2 &&
         ?e4 != ?e0 && ?e4 != ?e1 &&
         ?e4 != ?e2 && ?e4 != ?e3)
}
``` |
| fro_2-2 | ```
ASK WHERE {
  ?e0 {rel1} ?e2 .
  ?e1 {rel1} ?e2 .
  ?e3 {rel2} ?e2 .
  ?e4 {rel2} ?e2 .
  FILTER(?e0 != ?e1 && ?e1 != ?e2 &&
         ?e2 != ?e3 && ?e3 != ?e0 &&
         ?e0 != ?e2 && ?e1 != ?e2 &&
         ?e4 != ?e0 && ?e4 != ?e1 &&
         ?e4 != ?e2 && ?e4 != ?e3)
}
``` |
| | Continued on next page |

**Table 20 – continued from previous page**

| Pattern | SPARQL Query |
|---|---|
| rfo_2-2 | ```ASK WHERE {
  ?e2 {rel1} ?e0 .
  ?e2 {rel1} ?e1 .
  ?e2 {rel2} ?e3 .
  ?e2 {rel2} ?e4 .
  FILTER(?e0 != ?e1 && ?e1 != ?e2 &&
         ?e2 != ?e3 && ?e3 != ?e0 &&
         ?e0 != ?e2 && ?e1 != ?e2 &&
         ?e4 != ?e0 && ?e4 != ?e1 &&
         ?e4 != ?e2 && ?e4 != ?e3)
}``` |
| rro_2-2 | ```ASK WHERE {
  ?e2 {rel1} ?e0 .
  ?e2 {rel1} ?e1 .
  ?e3 {rel2} ?e2 .
  ?e4 {rel2} ?e2 .
  FILTER(?e0 != ?e1 && ?e1 != ?e2 &&
         ?e2 != ?e3 && ?e3 != ?e0 &&
         ?e0 != ?e2 && ?e1 != ?e2 &&
         ?e4 != ?e0 && ?e4 != ?e1 &&
         ?e4 != ?e2 && ?e4 != ?e3)
}``` |

