# OpenReview forum: "Graphlets as Building Blocks for Structural Vocabulary in Graph Foundation Models"
_ICLR.cc/2026/Conference — Submitted to ICLR 2026_

### Official Review · Reviewer_2ruj · 2025-10-17

**Soundness:** 2
**Presentation:** 1
**Contribution:** 2
**Rating:** 2
**Confidence:** 5

**Summary:**

This paper introduces ULTRA+, a knowledge graph foundation model that aims to improve ULTRA and MOTIF by replacing SPMM with pattern-matching with SPARQL and by reducing the high-arity pattern to positional-binary edges to use a single relation graph to compute the relation invariant. They further highlight the importance of open and closed paths when considered as graphlets, and with additional graphlets consideration, such as star-shaped graphs. Empirically, they report a marginal gain over ULTRA and MOTIF on zero-shot link prediction across 51 KGs.

**Strengths:**

- Pattern matching with SPARQL is a practical method that augments MOTIF and ULTRA, and it naturally supports open/closed distinctions, which is nice to see as this has been implemented beyond SPMM, which are quite limited.
- Binary version and summarization with positional binary edges are novel and a clean engineering trick.

**Weaknesses:**

**Major**:
- **Limited theoretical contribution**: The theorems are stated without proofs, which makes it hard to assess the core claims. In particular, a formal treatment of separation power (e.g., a WL-style characterization as in [1]) would substantially strengthen the work.
- **Open vs. closed paths claim**: MOTIF can theoretically distinguish open vs. closed paths if the motif family includes the relevant closed patterns (e.g., cycles). In that sense, the architectural difference emphasized for ULTRA+ appears to be a direct corollary to Theorem 6.4 in [1].
- **Runtime/efficiency claims**: The paper states that SPARQL-based construction is “computationally less demanding” than the SPMM kernel, but neither a complexity analysis nor an empirical runtime/memory study is provided.
- **Scope of architectural novelty**: Beyond the relation-graph construction, ULTRA+ appears architecturally the same as ULTRA.
- **Limited empirical evaluation** The Author does not empirically evaluate nor compare with TRIX [2], which can also implicitly count the homomorphisms. KG-ICL[3] is also not considered, which serves as a strong baseline. Additionally, it would be nice to see how each model variation can catch up with further fine-tuning, as the current gains over the existing model are small with no error bar or significance tests.
- **Explanation over empirical evaluation is unsubstantiated**: Notice that in the zero-shot setting, there is technically no difference between transductive and inductive dataset splits since the KGFM model does not observe the relation types or their distribution, regardless (they only know from the pre-training mix). Thus, the explanation of why ULTRA+ is worse than MOTIF on the transductive dataset is not justified. The author should instead discuss what the actual differences between these classes of datasets are, e.g., regarding graph statistics, to further justify their claim.

**Minor**:
- **Small Bug in codebase**: During training, the author first computes the relation representation from the relation encoder and then applies edge dropout in the training mode. This edge dropout during training might potentially change the relation graph constructions and thus yield different relation representations.
- **Presentation**: There are noticeable typos and inconsistencies in the paper; figures and tables are not properly adjusted.

[1] Huang, Xingyue, et al. "How Expressive are Knowledge Graph Foundation Models?." arXiv preprint arXiv:2502.13339 (2025).

[2] Zhang, Yucheng, et al. "TRIX: A more expressive model for zero-shot domain transfer in knowledge graphs." arXiv preprint arXiv:2502.19512 (2025).

[3] Cui, Yuanning, Zequn Sun, and Wei Hu. "A prompt-based knowledge graph foundation model for universal in-context reasoning." Advances in Neural Information Processing Systems 37 (2024): 7095-7124.

**Questions:**

- How does the notion of graphlet in ULTRA+ differ precisely from motifs in MOTIF?

- Can the author formally compare the separation power of positional-binary constructions with MOTIF or TRIX (e.g., via WL-style arguments or expressivity hierarchies)?

- Will the author include proofs for the stated theorems?

- Can the author provide runtime/memory analysis for SPARQL-based construction vs. SPMM (theoretical + empirical timing)?

- Does the author have an ablation isolating the effect of counting on relation graphs?

- Within the MOTIF framework, can the author add a triangle/cycle motif baseline to directly test the open vs. closed claim, and compare to ULTRA+ both empirically and theoretically?

---

> ### Author Response · Authors · 2025-11-21
>
> Thank you for your constructive feedback.
>
> **W1&Q2&Q3[Proofs of Theos & Expressiveness]** In response to the reviewer’s comment, we added proofs for Theorems 3.2 and 4.3 in Appendix A, to strengthen the theoretical foundations of our approach.
> Regarding a WL-style characterization of Ultra+’s expressive power, we agree it would be valuable. However, formally analyzing separation power via the WL test is nontrivial and requires substantial additional theoretical development. We will consider incorporating additional theoretical discussion in the final version of the paper.
>
>
> **W2 & Q1** Ultra+ directly mines open and closed paths via SPARQL, whereas motif methods based on sparse adjacency-matrix multiplication cannot, distinguish closed and open paths.
> If we enhance Motif to distinguish open vs. closed paths, it is only by using HBNet and a relation hypergraph. Ultra+, by contrast, uses positional binary relations, yielding a binary relation graph, and the NBFNet architecture.
> While Motif and Ultra share the same expressiveness with a 2-path vocabulary, extending to 3-paths or longer would require Ultra to adopt a relation hypergraph for Theorem 6.4 to hold, something Ultra+ does not do. Thus, the differences between Motif and Ultra+ are fundamental, not a simple consequence of Theorem 6.4.
>
>
> **W3 & Q4[Run Time Efficiency]** We provided a detailed comparison of the overall time complexity, runtime, and memory usage of Ultra+ (which relies on SPARQL ASK query for relation graph construction) versus Ultra and Motif (which rely on SPMM for relation graph construction) in Appendix E.
>
>
> **W4** We acknowledge that ULTRA+ may appear architecturally similar to ULTRA beyond relation-graph construction. However, this is not a limitation; our main contributions are as follows.
>
> (i) Query-based relation graph extraction: We propose a flexible SPARQL-based extraction method that efficiently identifies informative structures without relying on SPMM or other operations on the adjacency matrix. For more complex graphlets the computation of the relation graph adjacency matrix becomes computationally expensive if the operations needed are beyond multiplication of the adjacency matrix with $E_h$ or $E_t$ (which are obtained by scatter max over first node dimension and the last node dimension). As we wanted to distinguish between loops and open path (also rejecting path containing loops) we needed to employ the Equations 6&7 in Appendix E.3  which took orders of magnitude longer than the Query based extraction
>
> (ii) Closed and open relations: We incorporate both closed and open graphlets to capture a wider range of relational patterns and nuances.
>
> (iii) Binary relations: We represent graphlets as positional binary relations rather than n-ary relations.
>
> (iv) Model-agnostic design: Ultra+ is modular and can be integrated into any KG Foundation Model that uses relation graphs or structural vocabulary, making it broadly applicable and adaptable across GFM architectures.
> Although our work does not introduce a new model, it substantially strengthens how KGFMs capture and represent complex structural patterns.
>
>
> **W5[Baselines]** We would like to clarify that our experiments are specifically designed to evaluate the performance of KG Foundation Models, and we have compared our approach with two established baselines in this area, namely Ultra and Motif. Nevertheless, we are willing to consider additional evaluations with other baselines such as the TRIX and KG-ICL models.
>
>
> **W7[Dropout & Improvement]** We acknowledge the bug in our codebase and appreciate the reviewer's feedback. This can indeed lead to inconsistent relation relation representations, but only during training.  As dropout is disabled during inference this will not affect relation representation during Inference. However this may lead to lower convergence speed during training. In fact, the authors of Motif have already addressed this issue by recomputing the relation graph after applying edge dropout. However, this is computationally heavy since it involves a graphlet extraction and relation encoding, which are computationally expensive.
>
>
> Regarding the presentation of the paper, we apologize for the noticeable typos and inconsistencies. We will proofread the paper again carefully and adjusted the figures and tables properly in the final manuscript.
>
>
> **Q5** We do not have an ablation study that isolates the effect of counting on relation graphs.
>
>
> **Q6** Adding triangle/cycle graphlets into the MOTIF framework  using SPMM is computationally expensive, as it requires direct computation of triangles, which can be costly when using sparse matrix multiplications.
> Although it is possible to replace the SPMM with SPARQL ASK Query and keep the relation graph as a hypergraph (as Ultra+ is model-agnostic), leaving the HBNet intact, we believe that training the resulting MOTIF model and evaluating it on an extensive list of KGs is beyond the scope of this work.

---

> > ### Comment · Reviewer_2ruj · 2025-11-21
> >
> > W1&Q2&Q3.   Thanks for adding the proofs for Theorems 3.2 and 4.3. I still believe that if the paper's major claims are regarding expressivity, it is essential to show formally how and why the new model could be more expressive, i.e., distinguish all triplets that the other framework is able to distinguish while distinguishing some triplets that the other framework cannot. This does not necessarily need to go through WL-test: e.g., a formal statement similar to TRIX would suffice.  Additionally, can you claim or give an educated guess that ULTRA+ is more expressive than MOTIF under the same motif sets (graphlet set)?
> >
> > W2 & Q1. I agree that the SPARQL inclusion is valuable and the existing SPMM has limitations, which I have acknowledged in my strength sections. However, this is a practical constraint, not a theoretical one.  My W2 asked that since MOTIF is equipped with a proper motif (not the one shown in experiments, but the proposed theoretical framework) can distinguish open and closed paths by Theorem 6.4 of MOTIF. Does ULTRA+ follow a similar argument, assuming that the motifs are always binary, thus the constructed graph will always be binary? (Notice that the question does not rely on relational hypergraphs, so the argument of "only by using HCNet and a relational hypergraph" does not hold).
> >
> > Additionally, Q1 is not answered. Please clarify the necessity of a new definition, "graphlet," and how they are different from motifs.
> >
> > W3 & Q4 (runtime analysis). Thanks for providing the scalability analysis.  This is very useful.
> > Minor point: I have noticed multiple typos in appendix E. (line 906 representaitons -> representation. line 879 multiplicaiton -> multiplication, line 897 compputation -> computation)
> >
> > W7. Yes, I agree. This is why I believe a GPU and speed comparison would be only fair if both models are under the same conditions. Given that your additional scalability analysis only measures the construction time for the relation graph, I think this is not a concern here.
> >
> > W6. Explanation over empirical evaluation is unsubstantiated. Please address this comment. For example, the updated manuscripts still state that "The choice of relation hypergraph favors a precise structural learning of the KG itself. Thus, Motif outperforms all models on transductive zero-shot link prediction." This statement in line 458 does not hold, since Motif also conducts zero-shot link prediction, meaning that the model does not get access to the testing KG, thus the claim "precise structural learning" is not supported. This raised concern about whether the results discussion is meaningful at all.
> >
> > Q6. I acknowledge the difficulty of integrating new experiments. However, I still believe a fair comparison between Motif and Ultra+ with the same structural vocabulary is needed to evaluate ULTRA+'s performances and to isolate the effect of additional structural vocabulary, which helps us understand and directly compare the differences between binary positional encoding and relational hypergraph approaches.

---

> > > ### Author Response · Authors · 2025-12-04
> > >
> > > **W1&Q2&Q3**. We would like to thank the reviewer for the prompt feedback. Firstly, we would like to emphasize that the word 'expressive' or any of its derivatives were never explicitly used in the paper. Instead, we focused on the performance against the SOTA and the ability Ultra+ to identify robust structural invariance (formalized in Theorem 2).
> > > Following the reviewer's suggestion, we added closed 3-paths to Motif and created a synthetic closed 3-path graph (see Figure 5 in Section A.3). We then formally demonstrated that Motif is unable to distinguish between two of the three relations. This is directly related to its use of relation hypergraphs.
> > >
> > > **W2 & Q1**. In Section A.3, we demonstrated that considering open and closed paths as motifs in the Motif model framework does not necessarily augment the model's expressive power.
> > > Ultra+ is not limited to the graphlets used in the paper; it encompasses all types of graphlet. Introducing the concept of positional binaries (or m-aries) ensures that simple, complex or higher-order graphlets (or n-aries) can be mapped to a lower-arity graphlet; specifically, binaries. Therefore, Ultra+ always constructs binary relation graphs. We agreed with the reviewer that this has nothing to do with the GNN architecture; rather, it is intrinsically related to the choice between binary and n-ary edges.
> > >
> > > **Q1.** (i) Definition of graphlets and motifs in relation to network subgraphs.
> > > In 1, Graphlets are small, connected, non-isomorphic, induced subgraphs [1]
> > > In 2, network motifs are those patterns for which the probability P of appearing in a randomized network an equal or greater number of times than in the real network is lower than a cutoff value (here P 0.01) [2].
> > >
> > > We notice that motif is a special graphlet. We added these references to Section 3.3, where we defined the terms graphlet and motif.
> > >
> > > (ii) From a structural vocabulary perspective,
> > > MOTIF defines motifs as n-ary graphlets whose occurrences are collected as hyperedges in a lifted relation hypergraph. In Ultra+, we introduce the notion of a graphlet as a small subgraph pattern in the original KG together with a positional binary order that connects only the first and last relations of the pattern.
> > > Formally, a structural vocabulary 𝑉 is a collection of such graphlets together with a counting function 𝜔 over their equivalence classes. This abstraction separates what structural pattern we look for (the graphlet) from how it is encoded in the GFM (binary relation graph vs. hypergraph). A motif can be seen as one specific way of using such a graphlet: each occurrence becomes an n-ary hyperedge. Ultra+, in contrast, converts the same pattern into a binary edges between relations, which can be consumed by a standard relation-graph GNN (NBFNet). This is what allows Ultra+ to be model-agnostic: the same graphlet vocabulary can feed either a relation hypergraph encoder (MOTIF) or a binary relation graph encoder (ULTRA-style).
> > >
> > > (iii) From the perspective of constructing a relation graph, our model is completely different from MOTIF's. We find it more reasonable to use the term graphlet than motif. Overall, we can conclude that Ultra+ and Motif are genuinely different from each other.
> > >
> > > [1] Ribeiro, Pedro, et al. "A survey on subgraph counting: concepts, algorithms, and applications to network motifs and graphlets." ACM computing surveys (csur) 54.2 (2021): 1-36.
> > > [2] Milo, Ron, et al. "Network motifs: simple building blocks of complex networks." Science 298.5594 (2002): 824-827.
> > >
> > > **W3 & Q4**. We corrected the typos and proofread the paper.
> > >
> > > **W6**. When evaluating performance on transductive datasets, we included datasets suitable for relation and tail prediction, as well as datasets suitable for tail prediction only (e.g. FB15K237_10/20/50). In the first version, we accidentally displayed metrics relating to head and tail prediction for all datasets. We updated the tables with the correct values for all datasets. We also updated the argumentation in the results section of our work and added Results for the finetuned performance.
> > >
> > > **Q6**. In response to the reviewer's request to compare Ultra+ and Motif equipped with closed and open paths respectively, we investigated adding the changed vocabulary to MOTIF. This is possible but is beyond the scope of this work as we specifically investigate binary relation graphs.

---

### Official Review · Reviewer_kvqC · 2025-10-18

**Soundness:** 2
**Presentation:** 3
**Contribution:** 2
**Rating:** 4
**Confidence:** 4

**Summary:**

The authors propose using graphlets as structural tokens to establish a shared vocabulary for Graph Foundation Models (GFMs).
Then, the authors introduce a model-agnostic framework that extracts and encodes graphlets (2- and 3-paths, closed triangles, and star structures) to capture structural invariances across heterogeneous KGs. This graphlet-based vocabulary enables zero-shot generalization across unseen graphs. Evaluations on 51 diverse KGs show that incorporating graphlets as structural tokens significantly enhances performance for both inductive and transductive link prediction tasks.

**Strengths:**

I like the idea of treating graphlets as a structural vocabulary that parallels the tokenization principle in language models, offering a clean conceptual bridge between discrete and relational domains.
Using graphlets provides interpretable subgraph-level structures that can be intuitively linked to semantic or relational motifs in KGs.
The framework directly targets a key limitation in current Graph Foundation Models - their difficulty in transferring across unseen graphs.

**Weaknesses:**

- The use of graphlets as structural primitives is not entirely new; prior works in network science and graph representation learning have explored motif-based or subgraph-based encodings.

- The paper does not discuss the expressive power of the proposed graphlet-based vocabulary in relation to established graph isomorphism tests, such as the Weisfeiler–Lehman (WL) hierarchy. It remains unclear whether incorporating graphlets enhances the representational capacity beyond standard GNNs.

- Extracting graphlets at scale (especially 3-paths or larger motifs) can be computationally expensive for large or dense graphs.

**Questions:**

- Does the approach capture semantic relations beyond structural similarity? For instance, can similar structures with different relational meanings be disambiguated?

- How does the proposed framework compare against motif-based GNNs or subgraph isomorphism-based methods in terms of both efficiency and generalization?

- How does the proposed graphlet-based structural vocabulary relate to the Weisfeiler–Lehman (WL) in terms of expressive power?

---

> ### Author Response · Authors · 2025-11-21
>
> Thank you for your constructive feedback. In the following we address the Questions/weaknesses mentioned in the review:
>
>
> **R3W1[Novelty&Contribution]**
> We acknowledge that graphlets have been widely used as structural primitives in prior work. Our contribution lies in how we leverage them within knowledge GFMs.
>
> - Query-based relation graph extraction: We propose a flexible SPARQL-based extraction method that efficiently identifies informative structures without relying on SPMM.
> - Closed and open relations: We incorporate both closed and open graphlets to capture a wider range of relational patterns and nuances.
> - Binary relations: We represent graphlets as positional binary relations rather than traditional n-ary relations.
> - Model-agnostic design: Ultra+ is modular and can be integrated into any GMF that uses relation graphs or structural vocabulary, making it broadly applicable and adaptable across GFM architectures.
>
>
> **R3W2[Expressiveness]:**
> We appreciate the reviewer’s suggestion for a more rigorous theoretical analysis of Ultra+’s expressive power. However, formally analyzing separation power via the WL test is nontrivial and requires substantial additional theoretical understanding and development. We will consider incorporating additional theoretical discussion in the final version of the paper.
>
>
> **R3W3[Novelty Compare to Standard GNNs]:**
> We acknowledge that a formal proof of the representational benefits of graphlets over standard GNNs is still missing. However, our empirical results show clear performance gains when graphlets are incorporated into the structural vocabulary for link prediction. By enriching the relation graph, especially through distinguishing closed and open path-based graphlets, GNNs learn more informative entity and relation embeddings. This suggests that graphlets help capture extra relational structure beyond what standard neighborhood-based GNNs can model.
>
>
> **R3W4[Computing Power on Dense KGs]:**
> Extracting higher-order graphlets is computationally costly, but the performance gains of Ultra+ justify this trade-off. Our 2-path variant, Ultra+V2, is more efficient and remains highly competitive: as shown in Tables 1&5, it is the second-best model overall and the top performer on the FB15K237-{10, 20, 50} KGs, where increasing the number (10, 20, 50) means densifying the KG. This makes Ultra+V2 a strong lightweight alternative to Ultra+.
>
>
> **R3Q1[Semantics Awareness]:**
> Ultra+ is designed to capture structural properties of KGs, but we acknowledge that structure alone may be insufficient for disambiguating similar patterns that carry different relational meanings. Incorporating complementary information, such as relation-specific embeddings or the reasoning capabilities of large language models (LLMs), is a promising direction to address this limitation.
>
>
> **R3Q2[Subgraph- & Motif-based GNNs]:**
> We compared Ultra+ with Motif (motif-based GNNs) and Ultra in terms of efficiency and generalization. For generalization, we evaluated Ultra+ in a zero-shot setting across the datasets used in the Motif and Ultra papers. As shown in Tables 1, 5, 6, and 7, Ultra+ achieves state-of-the-art performance on many benchmarks, highlighting its strong generalization ability.
>
>
> **R3Q3[Expressiveness]:**
> We appreciate the reviewer’s suggestion for a more rigorous theoretical analysis of Ultra+’s expressive power. However, formally analyzing separation power via the WL test is nontrivial and requires substantial additional theoretical understanding and development. We will consider incorporating additional theoretical discussion in the final version of the paper.

---

> > ### Comment · Reviewer_kvqC · 2025-11-26
> >
> > Thank you for the detailed clarifications.
> >
> > - Novelty & Contribution of Graphlets: The clarifications strengthen the engineering contributions of Ultra+, particularly regarding query-based extraction, the distinction between closed/open graphlets, positional binary encoding, and the modular design. However, without a theoretical characterization, it remains unclear whether the graphlet-based vocabulary meaningfully increases expressive power over standard message-passing GNNs. That said, the added explanation does clarify how Ultra+ differs from purely motif-based GNNs.
> >
> > - Expressiveness: While I understand space and complexity constraints, the current paper lacks even a lightweight theoretical discussion of expressiveness (e.g., a connection to WL tests or other capacity arguments). This makes it difficult to fully assess the generality of the proposed graphlet-driven structural vocabulary.
> >
> > - Computational Cost: The authors acknowledge the cost of graphlet extraction and highlight Ultra+V2 as a more efficient alternative. This is reasonable, and the additional empirical evidence helps justify the trade-off. However, questions remain regarding scalability on very large KGs.
> >
> > - Semantic Awareness: The rebuttal notes that structural similarity alone cannot disambiguate semantically distinct relations and suggests combining graphlets with LLM-based or relation-specific embeddings. This is reasonable, but the limitation is significant in domains where structure is only weakly correlated with semantics and should be discussed more explicitly in the paper.
> >
> > Thus, my scores remain unchanged, though I acknowledge the contribution as potentially impactful for GFMs.

---

### Official Review · Reviewer_R1zb · 2025-10-27

**Soundness:** 3
**Presentation:** 2
**Contribution:** 2
**Rating:** 4
**Confidence:** 3

**Summary:**

The paper introduces a model-agnostic framework Ultra+ that builds a vocabulary of small graphlets to address the challenge of creating transferable representations in Graph Foundation Models (GFMs) for Knowledge Graphs (KGs), which lack a universal geometric structure. This sounds like a meaningful and promising research question, both for GFMs and KGs. However, the paper still faces some issues that need further improvement and clarification.

**Strengths:**

The problem studied in this paper is highly important and offers insightful implications for applying graph foundation models to knowledge graphs. Apart from some minor details, the overall writing of the thesis is professional. The experiments appear sufficiently comprehensive and generally sound, both in terms of dataset selection and comparisons with baseline methods. This paper seems to have sufficient theoretical support. There are insights into the improvement of the Ultra framework. Such research is worthy of appreciation and recognition.

**Weaknesses:**

From the perspective of graph foundation model frameworks, Ultra+ is an extension of the existing Ultra framework, and its novelty appears somewhat limited. I'm not very clear whether the graph foundation model framework used in knowledge graphs is relatively similar and unified, but it is clear that innovation at the model level is not the main contribution of this article.

While the paper provides very detailed definitions, it only includes two theorems, which are insufficient to robustly support the core argument. It would be beneficial to rigorously demonstrate the superiority of Ultra+ from perspectives such as expressive power, similar to what was done in paper [1].

Theoretically-driven improvements are certainly appreciated, and some theoretical ideas may already be integrated into the main text. However, the contributions should be emphasized more clearly. At present, it is difficult to fully grasp the theoretical innovations and the sources of Ultra+'s advantages as a new framework.

Some details need to be improved and clarified:

The description of graph foundation models on line 114 is outdated. It is now recognized that large language models represent a key branch of graph foundation models, extending beyond the scope of pre-trained GNNs[2, 3].

Lines 165-166: The “product function” notation $\eta \cdot \rho \cdot \eta$ is non-standard and can be confusing.

Are the concepts of graphlets and motifs first introduced in this paper? If not, they should be appropriately cited.

As a theorem, Theorem 3.2 requires a proof or a citation to the original work where it was first proposed. Similarly for Theorem 4.3.

Subfigures (d3) and (e3) in Figure 2 are identical.

“the query triple q(h, ?)” on line 294 and “the query (h, q, ?)” on line 303 are inconsistent. This discrepancy is confusing. The notation for a query should be unified and clearly defined.

A key claimed advantage of Ultra+ on lines 328-329 is its ability to discriminate between closed and open paths, unlike Motif. This point would be significantly strengthened by providing a concrete example illustrating this discrimination and linking it to theoretical results about expressive power.

[1]Xingyue Huang et al. How Expressive are Knowledge Graph Foundation Models? ICML, 2025.

[2]Jiawei Liu et al. Graph foundation models: Concepts, opportunities and challenges. IEEE Transactions on Pattern Analysis and Machine Intelligence, 2025.

[3]Zehong Wang et al. Graph foundation models: A comprehensive survey. arXiv preprint arXiv:2505.15116, 2025.

**Questions:**

"Ultra+ extends this approach by incorporating a richer set of graphlet-based pattern," how large is the size of this Graphlets in the specific implementation? If it is less than 5 as shown in Figure 2, how to "capture more complex and higher-order interactions between relations"

Will expanding the size of Graphlets expand the range of structural vocabulary and further enhance the generalization performance of GFMs?

---

> ### Author Response · Authors · 2025-11-21
>
> Thank you for your constructive feedback. In the following we address the Questions/weaknesses mentioned in the review:
>
> **R2W1& W3 [Novelty & Contributions]**
>  We would like to clarify that our main contribution lies not in proposing a new model architecture, but in enhancing the structural vocabulary used in KGFMs. The key contributions of Ultra+ can be summarized as follows.
>
> (i) Query-based relation graph extraction: We propose a flexible SPARQL-based extraction method that efficiently identifies informative structures without relying on SPMM.
>
> (ii) Closed and open relations: We incorporate both closed and open graphlets to capture a wider range of relational patterns and nuances.
>
> (iii) Binary relations: We represent graphlets as positional binary relations than traditional n-ary relations.
>
> (iv) Model-agnostic design: Ultra+ is modular and can be integrated into any Graph Foundation Model that uses relation graphs or structural vocabulary, making it broadly applicable and adaptable across GFM architectures.
> Although our work does not introduce a new model, it substantially strengthens how KGFMs capture and represent complex structural patterns.
>
>
> **R2W2[Expressiveness]**
> We appreciate the reviewer’s suggestion for a more rigorous theoretical analysis of Ultra+’s expressive power. However, formally analyzing separation power via the WL test is nontrivial and requires substantial additional theoretical understanding and development. We will consider incorporating additional theoretical discussion in the final version of the paper.
>
>
> **R2W4[Suggestions for Improvement]**
>  We thank the reviewer for the careful reading and for pointing out typos and ambiguities. Suggested improvements will be incorporated into the revised manuscript.
>
>
> **R2Q1[Size of Graphlets]**
>  In our implementation, path-based graphlets include up to 3 triplets, and star-shaped graphlets up to 4, as showed in Figure 2. Table 1 shows the number of graphlets in each structural vocabulary. While this captures useful structural patterns, modeling more complex interactions would require larger graphlets, which quickly become computationally expensive. Exploring more efficient mechanisms for capturing higher-order structures is an important direction for future work and is added to our conclusion.
>
>
> **R2Q2[Voc Size Boost Performance]**
>  Our experiments show that enlarging path-based graphlets increases the structural vocabulary and improves GFM performance. However, adding topology-based graphlets leads to performance degradation, indicating that simply using larger graphlets is not inherently beneficial. Effective vocabulary expansion requires selecting complementary graphlet types rather than indiscriminately increasing their size, highlighting the need for a deeper understanding of the underlying structural patterns.

---

### Official Review · Reviewer_KyYd · 2025-10-30

**Soundness:** 3
**Presentation:** 2
**Contribution:** 2
**Rating:** 4
**Confidence:** 4

**Summary:**

This paper proposes a Graph Foundation Model (GFM) framework based on graphlet structural vocabulary, designed for knowledge graph reasoning and zero-shot link prediction tasks. The authors argue that existing GFMs (such as Ultra and Motif) are limited in their ability to capture complex structural patterns, particularly due to their neglect of closed paths and higher-order structures. To address this, Ultra+ introduces a rich graphlet-based vocabulary that includes 2-path, 3-path, and star-shaped motifs to capture more robust structural invariances. Experiments conducted on 51 knowledge graph datasets demonstrate that Ultra+ consistently outperforms Ultra and Motif in both inductive and zero-shot reasoning settings.

**Strengths:**

Overall, the paper demonstrates strong innovation, with rigorous theoretical definitions, comprehensive experimental design, and good reproducibility, offering a new perspective on the role of structural vocabulary in GFMs.

**S1.** This paper presents an innovative framework Ultra+ which enhances graph structural modeling capability by introducing a graphlet-based structural vocabulary.


**S2.** On methodology, Ultra+ adopts a two-stage message passing mechanism that decouples relation graph learning from entity embedding learning, thereby achieving strong inductive and zero-shot generalization capabilities on unseen entities and relations.


**S3.** The experimental section covers more than 50 knowledge graph datasets, validating the model’s generality and superiority in inductive reasoning and zero-shot link prediction, and demonstrating consistent and robust improvements over models such as Ultra and Motif.

**Weaknesses:**

**W1.**  The improvements over Ultra and Motif remain somewhat ambiguous, lacking a clear mechanistic explanation—particularly regarding why the introduction of closed paths and higher-order graphlets leads to theoretical and performance gains. Moreover, the paper does not clearly justify why only closed and open 2-paths, 3-paths, and star-shaped graphlets are considered to achieve robust invariance.

**W2.**  GFMs should not be limited to KG; the paper lacks a discussion on the potential significance and generalizability of the Ultra+ framework when applied to other graph datasets, such as social networks or molecular graphs.

**W3.** The paper does not clearly explain how the proposed structural vocabulary is sampled, nor does it provide the corresponding complexity analysis.

**W4.** The paper lacks a notation table, and the numerous mathematical symbols used throughout, along with the unclear connections and roles between definitions and theorems, make the paper difficult to follow. It is recommended to include clarifying explanations or detailed proofs to improve readability.

**W5.** The experiments are not sufficiently comprehensive and lack comparative evaluations with the latest Graph Foundation Model baselines.

**Questions:**

See the weakness above.

---

> ### Author Response · Authors · 2025-11-21
> **We appreciate the comment from the reviewer. We added some more Theoretical Results, and clarified notation.**
>
> Thank you for your constructive feedback. In the following we address the Questions/weaknesses mentioned in the review:
>
> **R1W1.1[Proof of expressiveness]** We appreciate the reviewer’s suggestion for a more rigorous theoretical analysis of Ultra+’s expressive power. However, formally analyzing separation power via the WL test is nontrivial and requires substantial additional theoretical development. We will consider incorporating additional theoretical discussion in the final version of the paper.
>
> **R1W1.2[Choice of small-size graphlets]** In our implementation, path-based graphlets include up to 3 triplets, and star-shaped graphlets up to 4, as shown in Figure 2. Table 1 shows the number of graphlets in each structural vocabulary. While this captures useful structural patterns, modeling more complex interactions would require larger graphlets, which quickly becomes computationally expensive. Exploring more efficient mechanisms for capturing higher-order structures is an important direction for future work and is added to our conclusion.
>
> **R1W2[Graph Foundation Models beyond KGs]** While our work focuses on KGs, we agree that Ultra+ has the potential to generalize to other graph domains. Graphlets provide a transferable structural vocabulary applicable to settings such as social and molecular graphs. We plan to explore these broader applications in future work. However, Ultra+ can not be applied to uni-relational graphs where the relation graph would collapse to a single node.
>
> **R1W3.1[Structural Vocabulary]** We apologize for the earlier lack of clarity. Ultra+ builds its structural vocabulary by sampling all graphlets within each node’s 3-hop neighborhood. We limit graphlets to this size to balance computational cost with the ability to capture meaningful structural patterns. Fig 2. Shows the set of graphlets used in Ultra+.
>
> **R1W3.2[Time Complexity]** We provide a detailed comparison of the overall time complexity of Ultra+ (which relies on SPARQL ASK query for relation graph construction) versus Ultra (which relies on SPMM for relation graph construction) in the Appendix E. We also conducted an experimental evaluation of computation times for querying and SPMM-based relation graph computation using the vocabulary employed by our models (see Table 9 in Appendix E.2).
>
> **R1W4:[Notation Table & Proofs]** To address this concern, we will add a notation table that summarizes the key mathematical symbols and their meanings used throughout the paper. Additionally, we have provided detailed proofs of Theorems 3.2 & 4.3 in Appendix A., which includes step-by-step derivations and explanations of the key results.
>
> **R1W5[More Baselines]** We would like to clarify that our experiments are specifically designed to evaluate the performance of Knowledge Graph Foundation Models, and we have compared our approach against two established baselines in this area: Ultra and Motif. Nevertheless, we are willing to consider additional evaluations with other baselines such as the TRIX and KG-ICL models.

---

### Author Response · Authors · 2025-12-04

Upon reviewing our paper, we made several changes to address the concerns raised by the reviewers. These changes include:

1. **Proofreading and editing**: We thoroughly proofread the paper to correct typos and grammatical errors, ensuring that the text is clear and concise.
2. **Clarifying novelty and contributions**: We revised the Introduction by adding the paragraph “Our Contributions” to clearly articulate the novelty and contributions of our model, ULTRA+. Specifically, we emphasized that our aim is not to propose a new architecture, but rather to improve model performance and robustness by enhancing the structural vocabulary used in knowledge graph foundation models.
3. **Addressing comparisons with MOTIF**: We added references to clarify the similarities and differences between graphlets and motifs. We also noted that, unlike MOTIF, which uses network motif-based GNNs, our model uses positional binary graphlets to represent structural patterns.
4. **Additional appendices**: We added several sections to the appendix to provide additional information and address reviewer concerns, including:
Proofs of theorems: We provided proofs for the two theorems stated in the paper.
Notation table: We included a notation table to clarify the terminology used in the paper.
Complexity analysis: We conducted a complexity analysis to evaluate the computational efficiency of our model.
Runtime and memory usage analysis: We analyzed the runtime and memory usage of our model to assess its practicality and address scalaility.
Sparse matrix multiplication formulation: We formulated a sparse matrix multiplication approach to incorporate closed and open graphlets into the ULTRA model.
Theoretical proof of MOTIF's limitations: We provided a theoretical proof of the limitations of MOTIF, equipped with closed and open motifs, in distinguishing between two of the three relations forming a closed 3-paths.
Further results in the Additional Results section, showing the finetuned performance of our Method

We believe that these changes have significantly improved the clarity, accuracy, and completeness of our paper. We are confident that our revised paper meets the high standards of your journal and look forward to the opportunity to publish our work.

We thank the reviewers for their thorough reading and useful suggestions.

---

### Meta-Review · Area_Chair_4CUq · 2026-01-07

**Summary:**

The paper introduces Ultra+, a modification of the KGFM Ultra where relation graphs are built based off more structural patterns (graphlets) mined in the original KG.

Reviewers are rather skeptical about the work and highlight several important problems:
* Novelty and insufficient theoretical motivation (KyYd, R1zb, 2ruj) - the proposed method is very similar to the existing baseline (MOTIF) and there is no compelling formal proof whether Ultra+ is strictly more powerful or incomparable. Architecturally, the model itself is a minor modification of the original Ultra.
* Small practical gains, scalability issues, and lack of comparisons with other strong baselines (KyYd, 2ruj) - average 1% MRR difference across 50 graphs is marginal and is already behind other works like TRIX and KG-ICL.

During the rebuttal, the authors fixed presentation issues of the manuscript but failed to provide theoretical motivation of the work and more experimental evidence with respect to existing works. Ultimately, for a "model paper" without theoretical contributions, the achieved results are rather incremental and do not warrant a publication at ICLR. Therefore, I recommend a reject.

**Reviewer Concerns:**

* Novelty and theory: the authors repeatedly refused to provide any theoretical backing of the expressiveness comparison against MOTIF (in terms of WL hierarchies or similar)
* Scalability: the authors acknowledged the model going beyond 3-paths and 4-stars would be too computationally expensive
* More baselines: no new comparisons were added (albeit there is a plenty of literature in the field)

**Reviewer Scores:**

Two reviewers (kvqC and 2ruj) explicitly noted their scores remain unchanged.
I do not think the other two reviewers would have changed their score either.

---

### Decision · Program_Chairs · 2026-01-26

Reject